# Routing and Reasoned Evaluation with Large Language Models

Guiyao Tie [1][*]   Tianyao Luo [1][*]   Xueyang Zhou [1]   Chaoran Hu [1]   Yunhong He [2]
Junran Wu [1]   Yuanfan Yao [3]   Pan Zhou [1]   Lichao Sun [2]

## Abstract

Large language models (LLMs) are increasingly used to provide automated assessments. However, practical deployment faces three persistent challenges: heterogeneous reliability across models, substantial latency and token costs, and the absence of principled strategies for allocating evaluation resources. We introduce $R^2$Eval, a routing-aware automated assessment framework that combines difficulty-aware routing with reasoned evaluation signals to dynamically select evaluator models on a per-instance basis under explicit accuracy, latency, and cost constraints. Our study makes three contributions: 1) we construct six difficulty-aware datasets spanning both reasoning-intensive (mathematics, logic, code) and non-reasoning (knowledge, roleplay, writing) tasks. 2) we provide a systematic empirical analysis of how reasoning traces produced by different evaluator models correlate with assessment outcomes, revealing substantial variance and systematic mismatches across difficulty regimes. 3) we develop and evaluate both offline and online routing strategies that adaptively allocate evaluation queries. Experiments across 19 language models demonstrate that $R^2$Eval significantly reduces evaluation cost and latency while maintaining close alignment with human assessments. These results establish $R^2$Eval as a scalable and reliable framework for large-scale model evaluation.

## 1. Introduction

Large language models (LLMs) have rapidly advanced the frontier of artificial intelligence, enabling applications in reasoning, dialogue, programming, and creative writing (Achiam et al., 2023; Touvron et al., 2023; Team et al., 2023).

As these models are increasingly integrated into real-world systems, the demand for scalable, reliable, and cost-efficient evaluation has become pressing. Pairwise evaluation (i.e., comparing two candidate responses) is often used as a closer proxy for human preferences than pointwise scoring (Wang et al., 2023). However, manual evaluation is both costly and difficult to scale, motivating the adoption of automated judges: either LLMs or large reasoning models (LRMs) that produce explicit intermediate thinking.

Despite the promise of automated judging, substantial challenges remain. LLM-based judges suffer from systematic biases such as position bias, verbosity preference, and self-alignment tendencies (Dubois et al., 2023; Gallegos et al., 2024), while also exhibiting instability across repeated trials. Moreover, evaluating with stronger models introduces unavoidable accuracy–latency–cost trade-offs. Recent LRMs, which expose intermediate reasoning traces (Guo et al., 2025), offer new opportunities for understanding how judges reason, yet there remains limited systematic benchmarking of **LRM-as-a-Judge**, especially regarding how reasoning traces relate to final verdicts. Furthermore, while routing strategies have been explored for generation tasks (Chen et al., 2023), the problem of *routing judges*—selecting among heterogeneous judges under budget constraints—remains underexplored. These gaps motivate our central research question: ***How can we characterize and route LLMs and LRMs as automated judges, balancing accuracy, latency, and cost while remaining faithful to human preferences?***

To address these challenges, we introduce **$R^2$Eval**, a framework for benchmarking and routing automated judges. $R^2$Eval separates these two roles while making their connection explicit: the benchmark layer fixes the dataset, pairwise candidate responses, human gold labels, and per-judge measurements, whereas the routing layer is introduced afterward as a deployment policy over already benchmarked judges. This design preserves comparability of per-judge results while enabling practical judge allocation under explicit accuracy, latency, and token-cost objectives.

Our framework evaluates both LLM- and LRM-based judges across accuracy, latency, and cost. LRMs additionally provide thinking traces, which we use to characterize

---

[*]Equal contribution   [1]Huazhong University of Science and Technology [2]Lehigh University [3]ByteDance. Correspondence to: Pan Zhou <panzhou@hust.edu.cn>.

*Proceedings of the 43$^{rd}$ International Conference on Machine Learning*, Seoul, South Korea. PMLR 306, 2026. Copyright 2026 by the author(s).

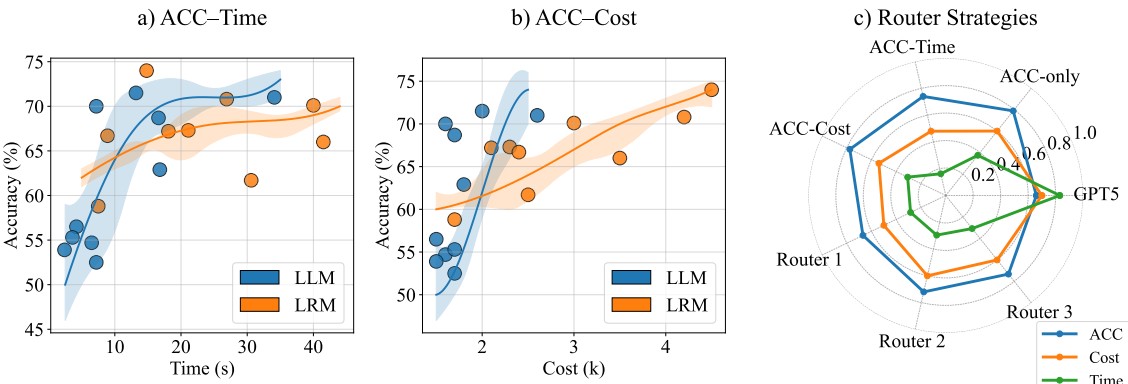

*Figure 1.* Representative results from R$^2$Eval. a) Accuracy–time trade-offs between LLM and LRM judges. b) Accuracy–cost trade-offs, where cost is measured by token usage. c) Accuracy–time–cost profiles of representative routing strategies under budget constraints. These views summarize the main deployment trade-offs rather than replacing the fixed per-judge benchmark results.

and diagnose judge behavior after execution, rather than as pre-routing inputs. Routing decisions rely only on low-cost signals available before invoking a judge, such as task/domain cues, difficulty proxies, and budget tags. As shown in Figure 1, LLM judges are generally faster and cheaper but exhibit higher variability, whereas LRMs can provide stronger and more interpretable judgments on structured reasoning tasks at greater computational cost. Our main contributions are as follows:

- **Benchmarking LRM-as-a-Judge.** We build a human-grounded pairwise benchmark protocol for evaluating LLM and LRM judges under a unified interface, covering six reasoning and non-reasoning domains with controlled standard and thinking-augmented variants.

- **Thinking–Verdict Diagnostics.** We analyze LRM thinking traces as diagnostic evidence for judge behavior, including confusion matrices between reasoning quality and verdict correctness and an error-type taxonomy that exposes mismatches such as "good thinking but wrong verdict."

- **Cost-Aware Judge Routing.** We design and evaluate offline and online judge-routing policies that expose accuracy–time–cost operating points, showing that routing can reduce evaluation overhead while preserving competitive agreement with human judgments under selected deployment settings.

## 2. R$^2$Eval

We present **R$^2$Eval**, a unified framework composed of three modules as shown in Fig. 2. The benchmark layer fixes the evaluation instances, pairwise responses, human gold labels, and per-judge measurements, while the routing layer is applied afterward as a deployment policy over already benchmarked judges. Thus, routing does not redefine the benchmark or alter per-judge evaluation results.

### 2.1. Preliminaries and Problem Setup

Each evaluation instance $x \in \mathcal{X}$ is paired with two candidate responses $(A, B)$ produced by distinct generators. [1] The judge must select a preference from $\mathcal{Y} = \{A, B, \text{Tie}\}$.

**Judge model.** A judge model $m$ implements a three-way probabilistic classifier

$$\mathbf{p}_m(x; A, B) = \left(p_m^A, p_m^B, p_m^{\text{Tie}}\right) \in \Delta^2, \qquad (1)$$

and outputs a verdict

$$\begin{aligned} \hat{y}_m(x) &= \arg\max_{y \in \mathcal{Y}} p_m^y(x; A, B), \\ c_m(x) &= \max_y p_m^y(x; A, B). \end{aligned} \qquad (2)$$

**Unified LLM/LRM interface.** LLMs return verdict-only outputs, while LRMs additionally produce explicit thinking traces. We use "thinking trace" as a unified term referring to any model-generated intermediate reasoning. Thinking traces provide a structured diagnostic signal for understanding judge behavior. Importantly, these traces are observed only after an LRM judge is invoked and are not used as pre-routing inputs in the current routing policies. This consistent API enables direct comparison between LLMs and LRMs and supports analysis of how intermediate reasoning relates to final verdicts.

### 2.2. Dataset Construction

**Pairwise task formation.** We curate two subdomains to probe complementary capabilities: reasoning (math, logic, code) and non-reasoning (knowledge, roleplay, writing). For each instance $x$, we obtain two responses $(A, B)$ from distinct generators. To mitigate position bias, we uniformly

---

[1] We use GPT-4o and Claude 3.5 as the two response generators. Additional generator-diversity analyses are provided in Appendix A.3.

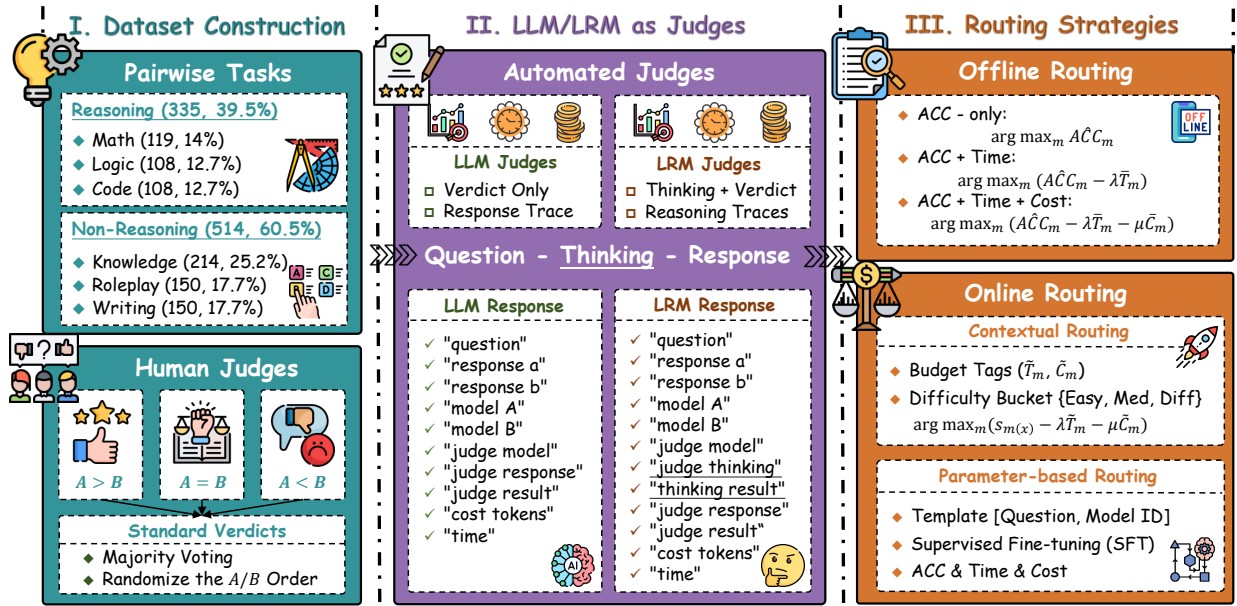

*Figure 2.* Overall pipeline of R²Eval. Stage I constructs a fixed pairwise benchmark with human preference labels. Stage II evaluates LLM and LRM judges under a unified interface; LRMs additionally produce thinking traces for post-hoc diagnostic analysis. Stage III applies routing as a deployment policy over the already benchmarked judge pool, using offline and online rankers to balance accuracy, latency, and token cost.

randomize the order of $(A, B)$, and log auxiliary features. We additionally provide both standard and thinking variants (including explicit reasoning traces from the generators). This controlled response protocol is used to compare judge behavior under a stable response distribution; its external-validity limitations and broader-generator robustness checks are discussed in Appendix A.3.

**Standard human verdicts.** Each triplet $(x, A, B)$ is annotated by $n = 7$ independent raters with labels $z_i(x) \in \mathcal{Y}$. The majority-vote label $y^{\text{MV}}(x) = \arg\max_{y \in \mathcal{Y}} \sum_{i=1}^{n} \mathbf{1}[z_i(x) = y]$ serves as the gold standard for evaluation.

### 2.3. LRM-as-a-Judge Benchmark

We extend the judge to LRMs with {verdict, confidence, rationale, thinking}. Unlike conventional LLMs, LRMs produce both verdicts and intermediate reasoning traces. This dual output opens the door to new questions: *Does high-quality reasoning necessarily yield correct verdicts? Can flawed reasoning still produce superficially correct answers?* By benchmarking LRMs across reasoning and non-reasoning domains, we establish a systematic evaluation of judge models that reason explicitly. This benchmark not only measures verdict accuracy but also uses thinking traces for post-hoc characterization of judge behavior, offering a diagnostic lens absent from prior verdict-only evaluations.

### 2.4. Routing Strategies

Given an instance $x$, the routing module must decide which judge $m \in \mathcal{M}$ to invoke. For each judge, we track its expected accuracy, time, and token cost and define the utility

$$U(m \,|\, x) = \underbrace{\mathbb{E}[\text{ACC}_m(x)]}_{\text{accuracy}} - \lambda \underbrace{\mathbb{E}[T_m(x)]}_{\text{time}} - \mu \underbrace{\mathbb{E}[C_m(x)]}_{\text{token cost}},$$
(3)

with deployment-dependent weights $\lambda, \mu \geq 0$. These coefficients specify an operating point on the accuracy–latency–cost frontier rather than universal constants; the values used in our main experiments are selected on the validation split and analyzed in Table 7. We also allow an abstention action $\perp$ (fallback-to-human) with utility $U(\perp \,|\, x)$. Figure 3 illustrates the overall routing framework: offline policies optimize global averages, while online strategies adaptively route per instance.

**Offline Routing (Instance-agnostic).** Using a domain-balanced validation split, we compute global accuracy $\hat{\text{ACC}}_m$, mean latency $\overline{T}_m$, and mean token cost $\overline{C}_m$ for all judges. These statistics define three fixed policies: (1) an accuracy-only policy: $\pi_{\text{A}}(x) = \arg\max_m \hat{\text{ACC}}_m$, (2) an accuracy–time policy: $\pi_{\text{AT}}(x) = \arg\max_m(\hat{\text{ACC}}_m - \lambda \overline{T}_m)$, (3) an accuracy–time–cost: $\pi_{\text{ATC}}(x) = \arg\max_m(\hat{\text{ACC}}_m - \lambda \overline{T}_m - \mu \overline{C}_m)$. Although easy to deploy and fully reproducible, these policies cannot incorporate instance-level variation such as domain, difficulty, or input complexity, and therefore serve as strong zero-overhead baselines for evaluating adaptive routing.

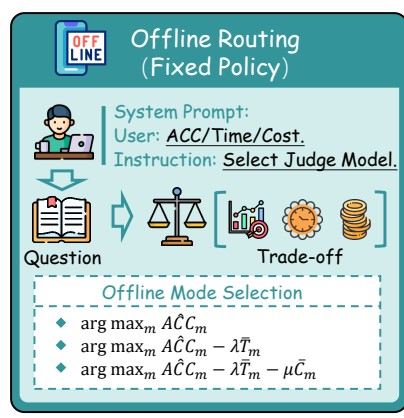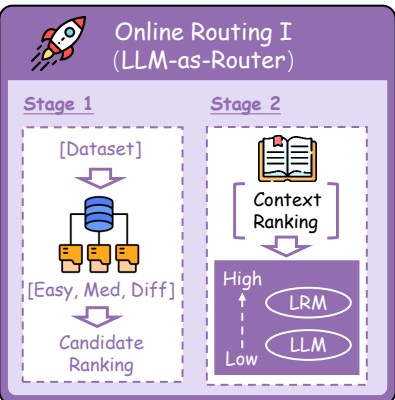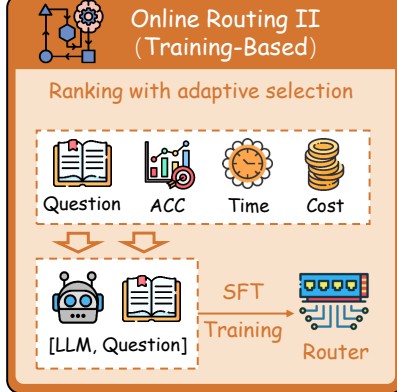

*Figure 3.* Routing-only view of R²Eval. Offline routing uses global validation statistics to produce fixed, instance-agnostic policies. Online routing adapts judge selection at the instance level using low-cost pre-decision features through either LLM-as-Router or a trainable router. Thinking traces are observed only after an LRM judge is invoked and are not used as router inputs in the current implementation.

**Online Routing I: Contextual (LLM-as-Router).** To enable adaptivity, we employ an LLM router. The router is prompted with compact features of $x$ (length, domain cues), a brief legend of each judge's historical strengths, and budget tags $(\tilde{T}_m, \tilde{C}_m)$. In our difficulty-aware variant, we first map $x$ to a difficulty-proxy bucket $b \in \{\text{Easy}, \text{Medium}, \text{Diff}\}$ using lightweight heuristics; we then restrict attention to a pre-defined model band $\mathcal{S}_b$ constructed from offline cost-aware rankings (low $\rightarrow$ high within the band). These buckets are used as routing proxies rather than an external ground-truth notion of difficulty. The router outputs scores $\{s_m(x)\}_{m \in \mathcal{S}_b}$ and we select

$$\pi_{\text{ctx}}(x) = \arg\max_{m \in \mathcal{S}_b} \left( s_m(x) - \lambda \tilde{T}_m - \mu \tilde{C}_m \right). \quad (4)$$

A confidence gate $\max_m s_m(x) < \tau$ triggers escalation within the band (low→high) or abstention $\bot$. This design couples global efficiency with local adaptivity, enabling budget-aware judge selection.

**Online Routing II: Training-based (Trainable Router).** We additionally develop a training-based router that learns routing decisions from data. The router is a lightweight 7B–8B model $r_\theta : \phi(x) \rightarrow \Delta^{|\mathcal{M}|+1}$, where $\phi(x)$ contains inexpensive input features such as the question text and the judge recommended by the offline utility-maximizing policy. Gold labels, difficulty annotations, and human preference signals are never used as router inputs, ensuring that routing decisions are not driven by label-derived information. Training data are constructed by pairing each instance $x$ with its offline-selected judge, treated as the supervisory target. More formally, let

$$m^*(x) = \arg\max_{m \in \mathcal{M}} \hat{U}(m \mid x) \quad (5)$$

denote the utility-maximizing judge assignment computed from training/validation utility estimates. The trainable

router is optimized with a cross-entropy objective:

$$\mathcal{L}(\theta) = - \sum_{x \in \mathcal{D}_{\text{train}}} \log r_\theta(m^*(x) \mid \phi(x)). \quad (6)$$

The router is trained to predict a distribution over all candidate judges that aligns with utility-maximizing choices, allowing it to depart from a single global fixed policy when the input features suggest a potential improvement in accuracy, time, or cost. The dataset is split into train, validation, and test by domain to prevent leakage of instance-level signals across splits. At inference time, the router processes a short input prompt and outputs a single routing decision. End-to-end latency and cost measurements, including router computation, are reported in Table 8. The training-based router therefore provides a scalable, low-cost mechanism for incorporating instance-specific information into routing policies while maintaining clear separation between routing features and human evaluation signals.

## 3. Experiment

### 3.1. Experiment Setup

**Dataset.** The **R²Eval** corpus supports systematic evaluation of automated judges under a unified pairwise setting. Each instance contains a prompt and two candidate responses $(A, B)$ generated by **GPT-4o** and **Claude 3.5**, following JudgeBench and FairEval to ensure stylistic diversity and avoid single-model bias. Human gold labels come from seven independent annotators. The dataset spans six domains—Math, Logic, Code, Knowledge, Roleplay, Writing—grouped into reasoning and non-reasoning. Difficulty buckets (EASY/MEDIUM/DIFF) are assigned by evaluating each item across a fixed panel of baseline judges and partitioning by average correctness, producing a baseline-panel-based stratification of relative hardness rather than an external, model-independent notion of difficulty. We discuss

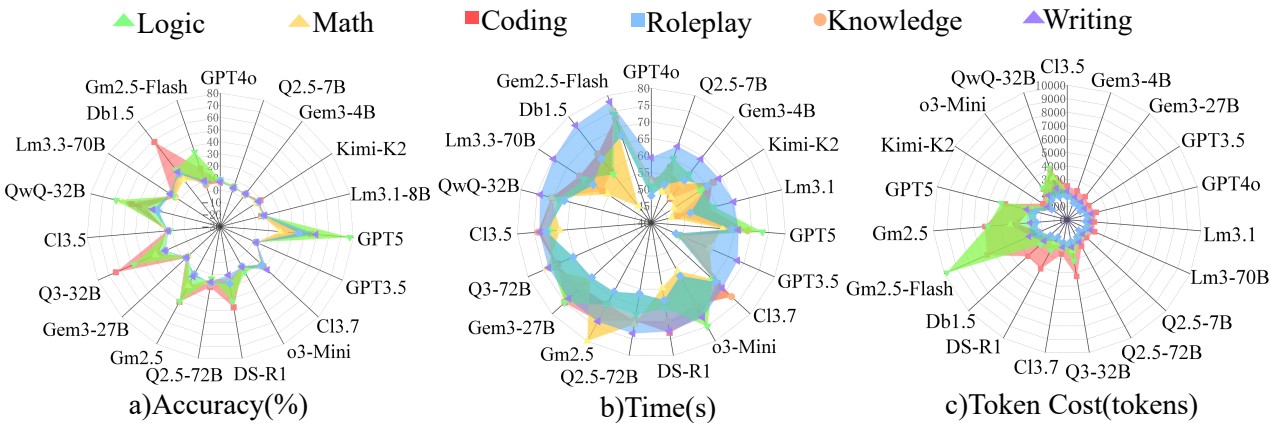

*Figure 4.* Aggregate radar analysis of accuracy–efficiency trade-offs across judges. The three panels summarize domain-level accuracy, latency, and token cost, respectively. For readability, model names are abbreviated as follows: Cl → Claude, Lm → Llama, Q → Qwen, Gm → Gemini, Gem → Gemma, DS → DeepSeek, Db → Doubao.

*Table 1.* Dataset statistics of the $\mathbf{R^2Eval}$. Difficulty buckets are baseline-panel-relative proxies rather than external ground-truth hardness labels.

| Category | Value |
|---|---|
| **Corpus** | |
| Total items | 849 |
| Reasoning : Non-reasoning | 335 : 514 |
| Candidate responses / item | 2 (A,B) |
| Total judgments | 5943 |
| **Domain distribution** | |
| Validation : Test | 96 : 753 |
| Math | 119 (14%) |
| Logic | 108 (12.7%) |
| Code | 108 (12.7%) |
| Knowledge | 214 (25.2%) |
| Roleplay | 150 (17.7%) |
| Writing | 150 (17.7%) |
| **Difficulty** | |
| Easy : Easy-Think | 323 : 239 |
| Medium : Medium-Think | 165 : 121 |
| Diff : Diff-Think | 175 : 83 |

the possible endogeneity of this design in Appendix A.4. For each item, we also include a THINK variant in which the original generators provide explicit reasoning traces. These paired versions allow controlled comparison of verdict-only judging and thinking-augmented judging. Full data provenance is provided in Appendix A.2.

**Models.** We evaluate 19 judge models, where 10 models are standard LLM judges that output a direct verdict without structured intermediate reasoning, and 9 models are LRMs (e.g., DeepSeek-R1, o3-mini) that generate explicit thinking traces before producing a final judgment. LRMs and LLMs are handled through a unified judge interface. To ensure comparability across models, all LLM judges are evaluated with temperature = 0 for deterministic verdicts.

LRM judges are evaluated using their recommended decoding temperatures to preserve reasoning stability. The 1024-token cap applies to the final verdict/rationale field used for standardized parsing and cost accounting, while LRM thinking traces are collected separately and are not truncated by this final-response cap.

### 3.2. Main Results

Table 2 reports accuracy across reasoning and non-reasoning domains. Three observations emerge. 1) LRMs obtain the strongest performance in reasoning domains: *Gemini-2.5-Flash* (74.90%) and *Gemini-2.5-Pro* (73.63%) lead in math and logic. Paired bootstrap tests (10k samples) yield narrow confidence intervals (average width 1.7 pp), supporting the stability of the aggregate comparisons. 2) In non-reasoning domains, compact instruction-following models remain competitive. *Claude-3.5-Sonnet* achieves the best writing and roleplay accuracy, highlighting that explicit multi-step reasoning is less beneficial when human preference is driven by tone, pragmatics, or stylistic alignment. 3) Domain specialization is evident: some models (e.g., *Gemma-3-27b-it*) excel in coding and logic but lag in subjective domains, whereas others (e.g., *o3-Mini*) score well in logic but struggle in math. Intermediate models such as *Qwen3-32B* and *DeepSeek-R1* show the lowest cross-domain variance, suggesting that robustness is not explained by scale alone. To contextualize variability in human preference, we compute inter-annotator agreement on the $R^2$Eval corpus (Table 4). Structured tasks such as math and logic yield higher agreement (Fleiss' $\kappa$ between 0.78 and 0.81), whereas roleplay and writing show substantially more nearties (21–23%), reflecting their inherent subjectivity. Additionally, the "Avg. Thinking Length" is measured in tokens (via the GPT-4o tokenizer). Explicit reasoning length correlates positively with accuracy on reasoning tasks ($r = 0.28$)

*Table 2.* Comparison of judge accuracy (%) across reasoning and non-reasoning tasks. Best results per column are in **bold**, second-best are underlined. Within each row, the maximum and minimum domain scores are highlighted in green and red, respectively.

| Model | Split | | Reasoning | | | | Non-Reasoning | | | | Overall |
|---|---|---|---|---|---|---|---|---|---|---|---|
| | Val | Test | Coding | Math | Logic | Avg | Knowledge | Roleplay | Writing | Avg | |
| **Baselines** | | | | | | | | | | | |
| Random Choice | 33.3 | 33.3 | 33.3 | 36.97 | 36.11 | 35.46 | 33.18 | 36.67 | 33.3 | 34.39 | 34.93 |
| Frequent Choice | 36.82 | 35.91 | 38.03 | 37.53 | 36.27 | 37.23 | 34.54 | 36.71 | 34.95 | 35.41 | 36.37 |
| Expert (Human & GPT4o) | **83.24** | **77.93** | - | - | - | - | - | - | - | - | - |
| **Large Language Models (without Thinking)** | | | | | | | | | | | |
| Llama-3.1-8B-Instruct | 52.08 | 49.34 | 48.15 | 46.22 | 59.26 | 51.21 | 57.48 | 52.00 | 65.33 | 58.27 | 54.74 |
| Llama-3.3-70B-Instruct | 59.38 | 56.17 | 57.41 | 55.46 | 63.89 | 58.92 | 65.42 | 60.67 | 74.67 | 66.92 | 62.25 |
| Qwen2.5-7B-Instruct | 62.50 | 58.96 | 57.41 | 49.58 | 60.19 | 55.73 | 56.07 | 52.00 | 64.00 | 57.36 | 56.88 |
| Qwen2.5-72B-Instruct | 65.62 | 62.33 | 69.44 | 70.59 | 67.59 | 69.21 | 69.63 | 61.33 | 73.33 | 68.10 | 68.65 |
| Qwen3-32B | 70.83 | 67.44 | 70.37 | 73.95 | 72.22 | 72.18 | 71.03 | 65.33 | 73.33 | 69.90 | 71.04 |
| Gemma-3-4b-it | 53.12 | 51.07 | 49.07 | 49.58 | 58.33 | 52.33 | 54.67 | 56.00 | 64.00 | 58.22 | 55.61 |
| Gemma-3-27b-it | 68.75 | 65.92 | 75.00 | 64.71 | 75.00 | 71.57 | 68.22 | 64.00 | 73.33 | 68.52 | 70.38 |
| GPT-3.5-Turbo-1106 | 54.17 | 51.68 | 49.07 | 58.82 | 49.07 | 52.32 | 50.47 | 48.00 | 68.00 | 55.49 | 53.57 |
| GPT4o | 45.83 | 43.11 | 51.85 | 52.10 | 50.93 | 51.63 | 52.80 | 48.00 | 59.33 | 53.38 | 52.50 |
| Claude-3.5-Sonnet | 71.88 | 68.72 | 73.14 | 67.23 | 70.37 | 70.25 | **73.83** | **71.33** | 73.33 | **72.83** | 71.71 |
| **Large Reasoning Models (with Thinking)** | | | | | | | | | | | |
| QwQ-32B | **73.96** | **70.43** | 70.37 | 70.59 | 68.52 | 69.83 | 70.56 | 66.67 | 74.00 | 70.41 | 70.45 |
| Kimi-K2-Instruct | 56.25 | 53.77 | 62.04 | 47.90 | 57.41 | 55.78 | 61.68 | 60.00 | 64.00 | 61.89 | 58.84 |
| Doubao-1.5-Thinking-Pro | 63.54 | 59.93 | 59.26 | 46.22 | 58.33 | 54.60 | 65.89 | 64.00 | 76.67 | 68.85 | 61.06 |
| DeepSeek-R1 | 70.83 | 67.08 | 73.15 | 59.66 | 65.74 | 66.18 | 69.63 | 63.33 | 72.67 | 68.54 | 67.03 |
| Claude-3.7-Sonnet | 68.75 | 65.62 | 66.67 | 64.71 | 64.81 | 65.40 | 72.43 | 66.00 | 68.67 | 69.03 | 67.55 |
| Gemini-2.5-Flash | 69.79 | 66.45 | 75.00 | 75.63 | 74.07 | **74.90** | 72.90 | 67.33 | **78.00** | 72.74 | **73.82** |
| Gemini-2.5-Pro | 66.04 | 63.25 | 73.15 | **80.00** | 67.74 | 73.63 | 67.44 | 63.33 | 73.33 | 68.03 | 70.83 |
| o3-Mini | 63.54 | 60.18 | 71.30 | 56.30 | **75.00** | 67.53 | 67.76 | 58.00 | 72.00 | 65.92 | 66.73 |
| GPT5 | 66.67 | 63.82 | 62.96 | 61.34 | 73.15 | 65.82 | 68.69 | 64.00 | 66.00 | 66.23 | 66.69 |

but shows negligible correlation on subjective tasks. Finally, temperature-sensitivity experiments indicate that standard LLMs behave most stably at temperature 0, whereas LRMs such as *DeepSeek-R1* perform slightly better at 0.5–0.6.

### 3.3. Time and Cost Latency Results

Figure 4 summarizes the accuracy–latency–cost trade-offs. LLMs form a low-cost, low-latency frontier but do not achieve the highest accuracy. LRMs incur substantially more tokens because of explicit reasoning traces; for example, *Gemini-2.5-Flash* produces longer thinking sequences (median 250 tokens), explaining its higher cost despite a 1024-token cap. Mid-sized models (*Qwen3-32B*) trace out a Pareto-efficient region, reducing latency by 35–50% relative to the strongest LRMs while staying within 2–3 pp of their accuracy. Variance profiles also differ: LLMs exhibit tight latency distributions, whereas multi-stage LRM reasoning induces broader spreads. These patterns further motivate routing mechanisms for accuracy-efficiency balancing.

### 3.4. Pipeline Demonstration

Figure 5 illustrates how R$^2$Eval integrates human annotation, automated judging, and routing into a coherent workflow. The pipeline is designed around two principles. First, a unified judge interface enables LLMs and LRMs to be evaluated under identical pairwise settings, while allowing LRMs to contribute structured thinking traces for interpretability. Second, routing is naturally layered: offline policies provide cost-aware global defaults, whereas online routing introduces instance-level adaptivity through contextual prompting or lightweight learned routers. This structure mirrors real-world deployment considerations, where judgments must remain reliable under budget constraints while leveraging explicit reasoning when beneficial. Importantly, the routing module is applied after per-judge benchmarking and does not change the fixed human labels or per-judge evaluation results.

### 3.5. Routing Strategy Comparison

Table 3 compares fixed-model baselines with offline and online routing strategies. The routing utilities use the validation-selected coefficients $\lambda = 0.05$ and $\mu = 0.01$;

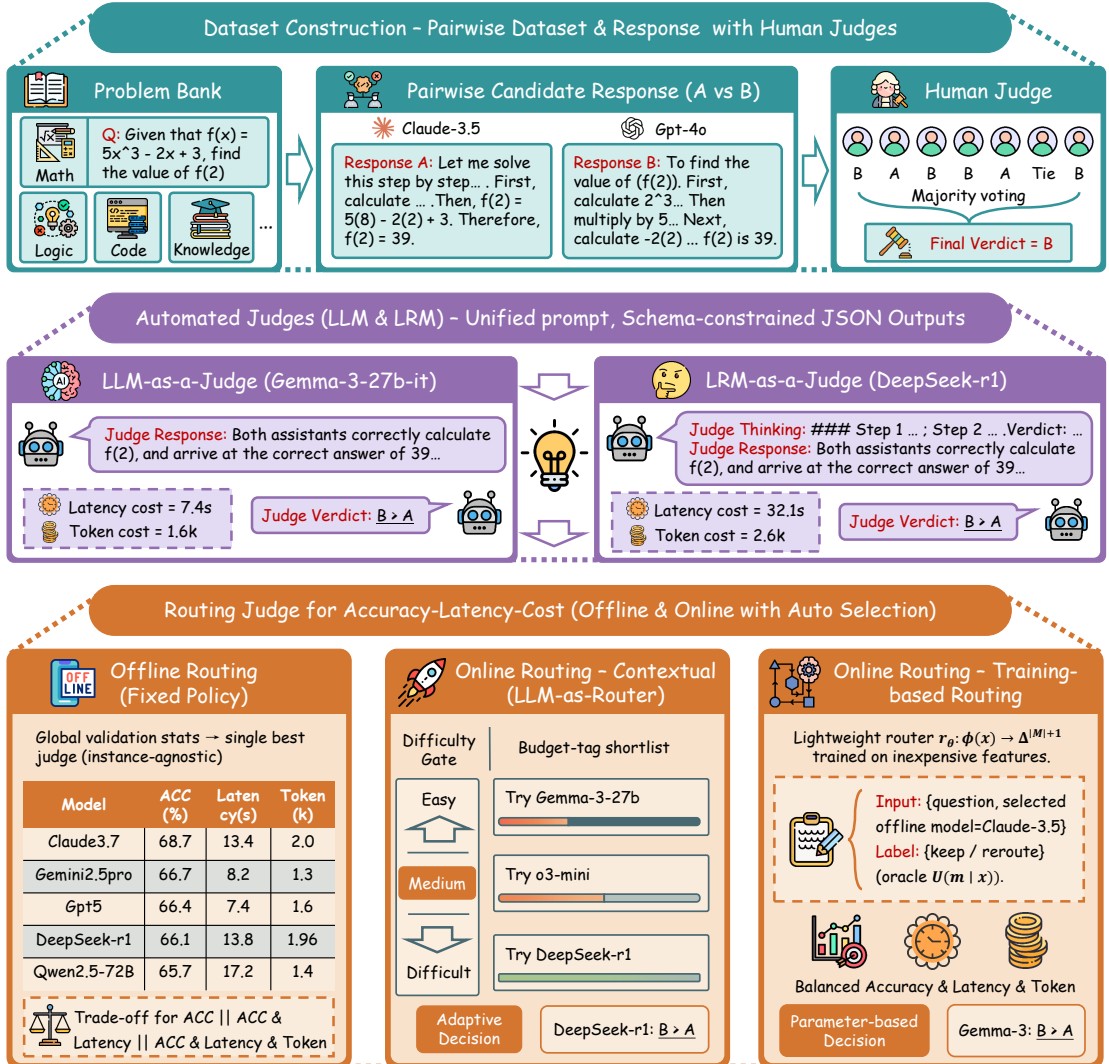

*Figure 5.* Example-level demonstration of the R$^2$Eval workflow. A pairwise task is first labeled by human annotators to obtain a fixed gold preference. LLM and LRM judges are then evaluated under the same pairwise interface, with LRMs additionally producing thinking traces for diagnostic analysis. Offline and online routing modules are applied afterward to select a judge under accuracy, latency, and token-cost constraints, without changing the fixed human labels or per-judge benchmark results.

a sensitivity sweep is reported in Table 7. Table 3 reports selected-judge execution cost and latency. We separately account for end-to-end overhead introduced by the router itself in Table 8. Offline heuristics provide strong global references: ACC-only maximizes accuracy by always selecting the strongest judge, ACC-Time minimizes latency by prioritizing compact LLMs, and ACC-Time-Cost trades off accuracy against both time and tokens. However, these policies ignore instance-level variation in domain and difficulty. The online routers address this limitation. Router 2 uses *Llama-3.1-8B* as a backbone, and Router 3 uses *Qwen2.5-7B* (Comprehensive implementation details are provided in Appendix A.1). These models were selected because they are substantially lighter than the strongest judges (30–70B or proprietary LRMs) yet retain enough capacity to detect structural

patterns in prompts and candidate responses. Router 2 tends to prefer judges that perform well on subjective tasks (e.g., knowledge, writing), reflecting the backbone's strength in fluency and stylistic cues. Router 3 performs best in Coding and Math, domains where shallow heuristics are insufficient and structural features (e.g., problem format, numerical operators) reliably map to stronger LRMs. This specialization yields distinct benefits. Router 2 achieves the highest non-reasoning accuracy (e.g., 77.2% in writing), whereas Router 3 achieves the highest accuracy in coding and math (74.6%, 76.3%). Both routers reduce selected-judge latency by 20–45% relative to ACC-only, and bootstrap tests indicate that Router 3 significantly exceeds ACC–Time in Coding and Math ($p < 0.05$). Router 1, which relies only on template-level cues, is efficient but insufficiently accurate.

*Table 3.* Quantitative comparison of routing strategies. Best accuracy is highlighted in green , best efficiency (lowest Time/Cost) in red . Time and cost report selected-judge execution; end-to-end router overhead is reported in Table 8.

| Strategy | Coding | | | Knowledge | | | Math | | | Reasoning | | | Roleplay | | | Writing | | | Average | | |
|---|---|---|---|---|---|---|---|---|---|---|---|---|---|---|---|---|---|---|---|---|---|
| | ACC | T | C | ACC | T | C | ACC | T | C | ACC | T | C | ACC | T | C | ACC | T | C | ACC | T | C |
| **Baseline** | | | | | | | | | | | | | | | | | | | | | |
| GPT5 (fixed) | 62.9 | 44.8 | 4.7 | 68.7 | 23.2 | 2.8 | 61.3 | 19.5 | 2.7 | 73.2 | 77.2 | 5.1 | 64.0 | 35.0 | 2.4 | 66.0 | 49.1 | 3.1 | 66.0 | 41.5 | 3.5 |
| **Offline Routing** | | | | | | | | | | | | | | | | | | | | | |
| ACC-only | 74.4 | 37.4 | 4.1 | 73.8 | 13.9 | 1.9 | 80.0 | 25.5 | 4.3 | 75.0 | 12.3 | 3.0 | 71.3 | 13.2 | 1.7 | 78.0 | 10.0 | 3.1 | 75.4 | 18.7 | 3.0 |
| ACC–Time | 63.0 | 8.3 | 2.1 | 72.9 | 6.2 | 2.3 | 75.6 | 8.9 | 3.0 | 75.0 | 7.0 | 1.6 | 67.3 | 8.2 | 2.3 | 78.0 | 10.0 | 3.1 | 71.9 | 8.1 | 2.4 |
| ACC–Time–Cost | 72.0 | 35.5 | 3.7 | 72.5 | 12.1 | 1.8 | 78.3 | 21.2 | 4.0 | 74.5 | 11.0 | 2.0 | 70.0 | 12.7 | 1.6 | 77.5 | 9.8 | 2.5 | 74.1 | 15.4 | 2.7 |
| **Online Routing** | | | | | | | | | | | | | | | | | | | | | |
| Router 1 | 68.8 | 8.8 | 2.7 | 56.3 | 12.6 | 1.7 | 68.6 | 29.2 | 5.3 | 70.0 | 12.4 | 2.3 | 62.5 | 13.5 | 1.7 | 75.0 | 8.4 | 1.2 | 66.9 | 14.2 | 2.5 |
| Router 2 (Llama) | 72.5 | 11.4 | 3.4 | 70.8 | 14.8 | 2.5 | 71.0 | 20.6 | 4.2 | 72.0 | 14.9 | 2.8 | 68.0 | 15.2 | 2.4 | 77.2 | 12.1 | 2.0 | 72.0 | 14.8 | 3.0 |
| Router 3 (Qwen) | 74.6 | 12.8 | 3.8 | 69.1 | 15.6 | 2.6 | 76.3 | 19.4 | 4.0 | 73.5 | 16.2 | 2.9 | 69.4 | 16.5 | 2.2 | 76.0 | 11.9 | 2.1 | 73.2 | 15.4 | 3.0 |

*Table 4.* Inter-annotator agreement, tie/near-tie rates, and difficulty statistics. Agreement uses Krippendorff's $\alpha$ and Fleiss' $\kappa$. Near-ties denote 4–3 or 3–3–1.

| Domain | $\alpha$ | $\kappa$ | Tie / Near-tie |
|---|---|---|---|
| Coding | 0.79 | 0.76 | 14.8% |
| Math | 0.83 | 0.81 | 11.2% |
| Logic | 0.82 | 0.78 | 12.6% |
| Knowledge | 0.76 | 0.72 | 18.4% |
| Roleplay | 0.74 | 0.70 | 21.7% |
| Writing | 0.71 | 0.68 | 23.3% |
| Overall | 0.78 | 0.74 | 17.0% |

**Why Train a Router?** While LLM-as-Router enables contextual routing, it remains relatively expensive and exhibits instability across domains and prompt variants. The trainable router offers a complementary design: it is 20–30× cheaper to evaluate, requires only lightweight features, and generalizes more robustly under distribution shift.

### 3.6. Thinking vs. Response: Error-Type Analysis

This analysis is diagnostic: thinking traces are observed only after an LRM judge is invoked and are not used as pre-routing inputs in the current routing policies. Figure 6 analyzes discrepancies between intermediate reasoning and final verdicts. We distinguish thinking (multi-step traces) from short rationales occasionally returned by LLMs; only the former represents structured reasoning sequences. To ensure reliability, a subset of the error labels and confusion categories was cross-checked by human annotators. 1) Thinking-level errors are dominated by omitted details (33.2%) and feature bias (30.6%), while logical inconsistencies are infrequent. 2) At the response level, omissions become less common, but reliance on incorrect criteria increases, indicating that verdict construction can compensate for missing steps yet still fail by misapplying evaluation rules. 3) Heatmaps reveal how reasoning quality aligns with verdict correctness. Objective domains such as math and logic exhibit strong TR-JR alignment (over 90%), suggesting that reasoning traces are more closely associated with final verdict correctness in these tasks. In contrast, subjective domains show larger mismatches: Knowledge has a considerable share of good thinking + wrong verdict (10.4%), indicating under-utilization of sound reasoning, whereas writing shows high flawed thinking + correct verdict (30.6%), reflecting reliance on stylistic or fluency cues rather than reasoning consistency. These patterns show that thinking traces are informative but not fully faithful: flawed reasoning can still yield acceptable verdicts when models rely on surface-level cues.

### 3.7. Difficulty-Aware Results

To examine how judges behave under varying task difficulty, we introduce a stratified split: non-reasoning tasks are partitioned into *Easy/Medium/Diff*, while reasoning tasks are grouped into *Easy-Think/Medium-Think/Diff-Think*. These buckets are baseline-panel-relative difficulty proxies rather than external ground-truth hardness labels. This design enables us to test whether stronger judges can resolve harder subsets that systematically defeat weaker ones. As shown in Table 5, reasoning tasks with explicit thinking traces exhibit sharper stratification than non-reasoning tasks. Nearly half of the *Medium-Think* items remain unresolved when falling back to *Easy-Think* (46.4%), and over a quarter of *Diff-Think* cases remain unsolved even when stepping down to the easiest bucket (25.9%). In contrast, non-reasoning tasks display milder separation, with unresolved rates concentrated in the 23–34% range. These findings yield two insights. 1) Reasoning-intensive domains naturally amplify difficulty gaps, validating the importance of difficulty-aware benchmarks for probing robustness. 2) Unresolved transitions provide actionable signals for routing: high-difficulty items should be escalated to stronger (but costlier) judges, while easier subsets can be reliably delegated to cheaper ones.

*Table 5.* Difficulty-aware split with cross-bucket resolvability. Each row reports bucket sizes (count; proportion within category) and unresolved counts (absolute; relative to the higher bucket). Difficulty buckets are non-disjoint to evaluate stability across boundaries; specifically, the Medium and Difficult sets incorporate overlapping samples from the Easy and Medium tiers, respectively.

| Category | Bucket size (count; % within category) | | | Cross-bucket Unresolved (n; rate) | | | Total (share) |
| --- | --- | --- | --- | --- | --- | --- | --- |
| | Easy | Medium | Difficult | Difficult $\not\to$ Medium | Difficult $\not\to$ Easy | Medium $\not\to$ Easy | |
| **Non-reasoning** | 323 (48.7%) | 165 (24.9%) | 175 (26.4%) | 41 (23.4%) | 60 (34.3%) | 52 (31.5%) | 663 (60.0%) |
| **Reasoning** | 239 (54.0%) | 121 (27.3%) | 83 (18.7%) | 31 (27.7%) | 29 (25.9%) | 52 (46.4%) | 443 (40.0%) |

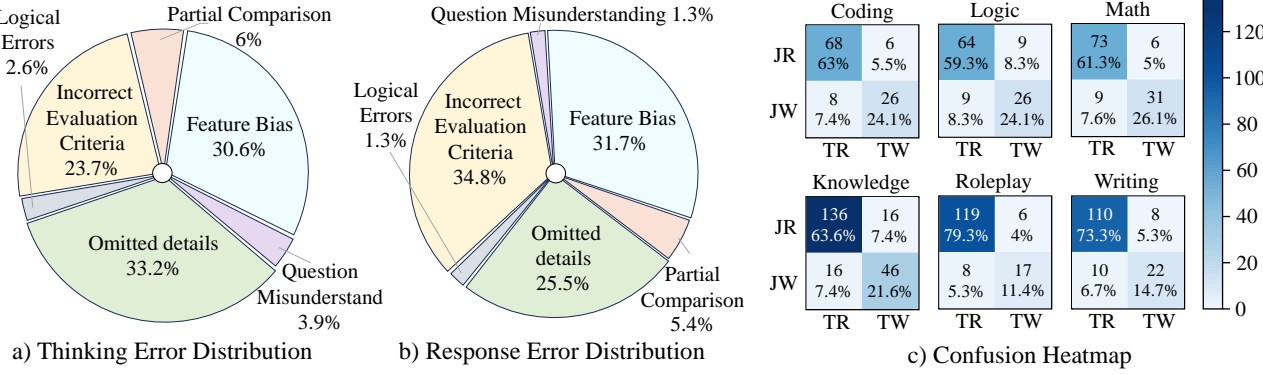

a) Thinking Error Distribution     b) Response Error Distribution     c) Confusion Heatmap

*Figure 6.* Diagnostic analysis of reasoning-level and verdict-level errors. Left: distribution of thinking-trace errors. Middle: distribution of response-level evaluation errors. Right: domain-wise heatmaps showing alignment or mismatch between reasoning quality and final verdict correctness, where TR/TW denote thinking-right/thinking-wrong and JR/JW denote judge-right/judge-wrong. Percentages are diagnostic proportions within analyzed categories, not global estimates of wasted reasoning over the full dataset.

Consistent with this view, Table 5 shows that unresolved transitions provide actionable signals for difficulty-aware judge allocation.

## 4. Related Work

**LLMs as Judges.** LLMs have been used as automated evaluators in MT-Bench and Chatbot Arena (Zheng et al., 2023), but exhibit systematic biases—position, verbosity, and self-preference (Chen et al., 2024; Shi et al., 2025; Wataoka et al., 2024; Li et al., 2025a). Broader analyses highlight instability and inconsistency (Li et al., 2025b; Anghel et al., 2025; Shi et al., 2024), suggesting that reliable evaluation requires multiple judges and principled routing rather than a single fixed evaluator.

**Reasoning Models and Thinking Traces.** LRMs provide explicit reasoning, yet prior work shows such traces may be helpful or misleading (Qi et al., 2025; Shojaee et al., 2025). Recent LRMs (e.g., DeepSeek-R1 (Guo et al., 2025), o3/o3-mini (Lai et al., 2025)) emphasize systematic reasoning, but no study evaluates LRM-as-a-Judge (Xu et al., 2025; Fan et al., 2025; Li et al., 2025c; Tie et al., 2025; Zheng et al., 2024; Duan et al., 2024; Zhu et al., 2022). Our benchmark addresses this via domain-wise confusion analysis and an error taxonomy for both thinking and judgments.

**Routing and Model Selection.** Routing in generation tasks has been explored through cascades, preference-trained routers (Li et al., 2023), and efficient verification

(Narasimhan et al., 2024). Judge routing raises additional challenges: heterogeneous biases (Trivedi et al., 2024; Sun et al., 2025; Ye & Ng, 2024), calibrated abstention (Jung et al., 2024), and reasoning-quality effects in LRMs (Chen et al., 2025; Xu et al., 2025; Fan et al., 2025; Li et al., 2025c; Tie et al., 2025; Zheng et al., 2024; Duan et al., 2024; Zhu et al., 2022). We study these factors jointly by routing across LLM and LRM judges with explicit accuracy–latency–cost considerations.

## 5. Conclusion

We presented **R²Eval**, a unified framework for evaluating and routing automated judges. Our results show that while LRMs yield more stable judgments on reasoning tasks, their explicit thinking incurs substantial cost, and correct reasoning does not always guarantee correct decisions, underscoring the need for reasoning-aware calibration. By stratifying instances by difficulty, R²Eval enables effective routing that assigns weaker judges to easy cases and reserves stronger LRMs for hard ones. As a result, routing exposes practical accuracy–latency–cost operating points and can reduce selected-judge evaluation overhead while preserving competitive agreement with human judgments under the evaluated settings. These findings offer a practical foundation for cost-aware and reliable automated evaluation.

## Impact Statement

This work introduces R$^2$Eval, a unified benchmarking and routing framework for automated LLM- and LRM-based judges under explicit accuracy, latency, and cost constraints. By systematically analyzing the relationship between reasoning traces and judgment correctness, R$^2$Eval reveals that explicit reasoning does not uniformly translate into reliable evaluation, highlighting the necessity of reasoning-aware calibration. The proposed difficulty-aware datasets and adaptive routing strategies provide practical mechanisms for allocating evaluation resources efficiently while maintaining alignment with human preferences. Our findings offer empirical foundations for scalable, cost-conscious, and interpretable automated evaluation, contributing to more reliable deployment of LLM-based systems in real-world assessment pipelines.

## Acknowledgments

This work is supported by the National Key R & D Program of China under No. 2024YFF0618700 and the National Natural Science Foundation of China (NSFC) under grant No. 62476107.

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

# A. Additional Details

## A.1. Implementation Details of Trainable Routers

**Training Objective and Data.** The trainable routers are implemented as sequence-to-sequence classifiers using the `Llama-3.1-8B-Instruct` backbone for Router 2 and the `Qwen2.5-7B-Instruct` backbone for Router 3. Given an input feature representation $\phi(x)$, the router predicts a distribution over candidate judges:

$$r_\theta : \phi(x) \to \Delta^{|\mathcal{M}|+1}. \tag{7}$$

For each training instance, we compute a utility-maximizing judge assignment:

$$m^*(x) = \arg \max_{m \in \mathcal{M}} \hat{U}(m \mid x), \tag{8}$$

where $\hat{U}(m \mid x)$ is computed from validation-derived accuracy, latency, and token-cost estimates. The router is trained with a cross-entropy objective:

$$\mathcal{L}(\theta) = - \sum_{x \in \mathcal{D}_{\text{train}}} \log r_\theta(m^*(x) \mid \phi(x)). \tag{9}$$

The router inputs include inexpensive pre-decision features such as question text, coarse domain cues, input length, and the judge recommended by the fixed utility policy. Human gold labels, difficulty annotations, and human preference signals are not used as router inputs.

**Hyperparameters and Environment.** Table 6 summarizes the hyperparameter settings used for fine-tuning.

All routers were trained on a single node equipped with 4×NVIDIA A100 (80GB) GPUs. At inference time, the router outputs a single judge-selection decision. We report both selected-judge cost and end-to-end cost including routing overhead in Table 8.

**Utility Weight Calibration.** For the quantitative comparisons in Table 3, we use $\lambda = 0.05$ for latency and $\mu = 0.01$ for token cost. These coefficients are deployment-dependent operating-point parameters rather than universal constants. They were selected on the validation split to balance accuracy, latency, and token cost. Table 7 reports a sensitivity sweep over representative coefficient settings.

**End-to-End Routing Overhead.** The main routing table reports selected-judge accuracy, latency, and token cost. For deployment, online routing also incurs routing-step overhead. Table 8 separates selected-judge execution from router overhead. Offline routing has no per-query routing overhead after validation statistics are fixed. Prompted LLM-as-Router introduces additional latency and token cost, whereas the trainable router remains closer to the selected-judge execution cost.

## A.2. Dataset Composition

$R^2$Eval differs from prior judge-oriented datasets (e.g., MT-Bench, FairEval, JudgeBench) in three important ways. First, it spans six domains across both reasoning and non-reasoning tasks, forming a broader evaluation space than existing judge benchmarks, which typically focus on only two or three domains. Second, $R^2$Eval provides paired *Think/Non-Think* variants for every item, enabling controlled analyses of explicit reasoning traces—an aspect absent in earlier datasets. Third, we adopt a dual-source response-generation protocol using GPT-4o and Claude 3.5, following the convention of prior judge studies while ensuring stylistic diversity and reducing model-specific bias. These design choices make $R^2$Eval a difficulty-aware, reasoning-augmented benchmark tailored specifically for studying both LLM and LRM judges.

Table 9 presents a consolidated summary of the **$R^2$Eval** corpus, including data provenance, domain definitions, pairwise construction, and human annotation metadata. This complements the high-level description in the main text and addresses reviewer questions regarding dataset sources, difficulty assignment, and thinking–response variants.

**Data provenance and coverage.** $R^2$Eval aggregates 849 pairwise instances drawn from twelve publicly available datasets (e.g., MT-Bench, FairEval, JudgeBench, APPS, MathQA). These sources span a broad spectrum of reasoning skills (symbolic logic, mathematical derivation, code correctness) and non-reasoning capabilities (factual knowledge, roleplay consistency, stylistic writing). Compared to prior judge benchmarks (e.g., MT-Bench, JudgeBench), $R^2$Eval covers *more domains*, *more difficulty diversity*, and includes *explicit reasoning variants*, enabling multi-dimensional analysis.

**Pairwise construction aligned with judge literature.** For each task, we generate two heterogeneous responses $(A, B)$ using **GPT-4o** and **Claude 3.5**, following established practice in JudgeBench and FairEval. This design ensures stylistic variety and avoids same-model correlations, yielding more realistic and challenging pairwise comparisons for automated judges.

**Difficulty-aware stratification.** $R^2$Eval uses a baseline-panel-based difficulty stratification. We evaluate each instance with a fixed panel of baseline judges and assign Easy/Medium/Diff buckets according to average correctness. These buckets should be interpreted as a reproducible, model-behavior-derived proxy for relative hardness, rather than an external or model-independent ground-truth notion of difficulty. This design supports controlled analysis of how judge behavior changes across coarse difficulty regimes.

Table 6. Hyperparameters for router fine-tuning.

| Hyperparameter | Value |
|---|---|
| Learning Rate | $1 \times 10^{-5}$ |
| Optimizer | AdamW ($\beta_1 = 0.9, \beta_2 = 0.95$) |
| Warmup Ratio | 0.05 |
| Weight Decay | 0.1 |

Table 7. Validation-based utility calibration for routing coefficients. Cost is measured in 1k tokens.

| $\lambda$ | $\mu$ | Router Acc. | Latency (s) | Cost |
|---|---|---|---|---|
| 0.00 | 0.000 | 79.1 | 5.8 | 8.7 |
| 0.02 | 0.005 | 78.6 | 4.6 | 6.4 |
| 0.05 | 0.010 | 77.8 | 3.1 | 4.2 |
| 0.08 | 0.020 | 76.9 | 2.4 | 3.1 |

Table 8. End-to-end routing overhead accounting. Latency is reported in seconds, and cost is reported in 1k tokens.

| Method | Acc. | Judge Lat. | Router Lat. | Total Lat. | Judge Cost | Router Cost | Total Cost |
|---|---|---|---|---|---|---|---|
| Offline ACC-Time-Cost | 74.1 | 15.4 | 0.00 | 15.40 | 2.7 | 0.00 | 2.70 |
| Online Router I | 72.0 | 14.8 | 0.30 | 15.10 | 3.0 | 0.20 | 3.20 |
| Online Router II | 73.2 | 15.4 | 0.05 | 15.45 | 3.0 | 0.03 | 3.03 |

Table 9. Comprehensive composition of the $R^2$Eval dataset. In the main benchmark, final pairwise responses $(A, B)$ are standardized to a controlled dual-generator protocol using GPT-4o and Claude 3.5.

| Domain | Total | Representative Source Datasets |
|---|---|---|
| Math | 119 | MT-Bench; FairEval; JudgeBench; MathQA; MiniF2F |
| Logic | 108 | ProverQA; MT-Bench; JudgeBench |
| Code | 108 | APPS; MT-Bench; FairEval; JudgeBench; LeetCode Hard |
| Knowledge | 214 | FairEval; JudgeBench; LLMEval2 |
| Roleplay | 150 | MT-Bench; FairEval; role_play_instruction |
| Writing | 150 | MT-Bench; FairEval; writing-1k |
| Total | 849 | Six domains with standard and thinking-augmented variants |

**Thinking and non-thinking variants.** For all items, we provide a parallel THINK version where response generators include explicit reasoning traces. This enables controlled experiments comparing verdict-only judges with LRM judges that consume intermediate thoughts. The ability to jointly evaluate verdict correctness, reasoning quality, and thinking–verdict mismatches is a distinctive contribution of $R^2$Eval and addresses gaps noted by reviewers (e.g., the lack of LRM-as-judge benchmarks).

**Human gold annotation and reliability.** Each instance is rated by seven independent annotators. Beyond majority vote, we record rater distributions and compute two reliability metrics Fleiss' $\kappa$ and Krippendorff's $\alpha$. These metadata support upper-bound estimation for routing and allow analysis of near-ties, annotator disagreement, and judge performance on ambiguous cases.

Together, these properties make $R^2$Eval a *multi-perspective*, *difficulty-aware*, and *reasoning-sensitive* corpus, designed specifically for studying LLM/LRM judges under accuracy–latency–cost trade-offs.

### A.3. Generator Robustness and Family-Overlap Analysis

The main $R^2$Eval benchmark uses GPT-4o and Claude 3.5 as a controlled dual-generator response protocol. This choice reduces confounding from heterogeneous generation pipelines and keeps pairwise comparisons non-trivial. However, it may raise concerns about response diversity and potential family-level preference effects. To examine whether the main findings depend on this controlled generator choice, we conduct two robustness analyses.

**Broader generator distribution.** We expand the response pool by adding strong open-source generators and mixed generator pairs. Table 10 shows that absolute accuracy decreases slightly under broader generator distributions, but the main trends remain stable.

*Table 10.* Broader-generator robustness analysis.

| Setting | Best Reasoning Judge | Best Non-Reasoning Judge | Router | Latency (s) |
|---|---|---|---|---|
| GPT-4o + Claude-3.5 | 78.6 | 75.9 | 77.8 | 3.1 |
| + Strong–Open pairs | 77.5 | 74.9 | 77.0 | 3.0 |
| + All generators mixed | 76.9 | 74.2 | 76.4 | 3.2 |

**Family-overlap split.** We further divide examples according to whether the judge family overlaps with either response generator family. Table 11 shows that the ranking of strong judges remains stable under the non-overlap split.

### A.4. Difficulty Stratification and Endogeneity

The difficulty buckets in $R^2$Eval are derived from average correctness across a fixed baseline judge panel. This design provides a practical and reproducible notion of relative hardness, but it is not an external ground-truth definition of difficulty. Because the buckets are derived from model behavior, they may inherit the biases and blind spots of the baseline panel.

We therefore interpret Easy/Medium/Diff buckets as baseline-panel-relative stratification rather than model-independent difficulty labels. Possible alternatives include disjoint baseline panels, leave-one-model-out difficulty estimation, human-agreement-based hardness proxies, and hybrid approaches that combine model correctness with annotator disagreement.

### A.5. Statistical Robustness under Coarser Groupings

Fine-grained slicing by domain, difficulty, and thinking condition can reduce statistical power. To avoid over-interpreting the smallest bins, we additionally evaluate the main conclusions under coarser groupings.

These results indicate that the main conclusions do not rely solely on the smallest fine-grained bins. Nevertheless, domain-level and difficulty-level analyses should be interpreted as diagnostic rather than definitive when the number of items is small.

### A.6. Additional Routing Baseline Comparisons

To contextualize the proposed routing strategies, we adapt representative routing and model-selection baselines to the judge-allocation setting. These baselines include RouteLLM, Prompt-to-Leaderboard, and ThinkSwitcher-style selection. All methods are evaluated over the same candidate judge pool where possible.

These comparisons are not intended to claim that general-purpose generation routers directly solve judge routing. Rather, they show that adapting existing routing ideas to judge allocation is non-trivial, and that explicitly modeling

judge accuracy, latency, and token cost provides practical benefits.

### A.7. Cross-Domain Routing Generalization

Offline routing policies are strong on the validation distribution because they directly optimize global statistics. Online routers, by contrast, are designed to incorporate instance-level features and can be more stable under transfer.

These results clarify the role of online routing: it does not necessarily dominate offline routing in every in-domain setting, but it provides a more adaptive policy that can reduce utility degradation under transfer.

### A.8. Error Taxonomy and Labeling Protocol

For the thinking–verdict analysis, we use a non-exhaustive taxonomy of reasoning and response-level errors. The goal is not to assign a complete causal explanation to every failure, but to identify the dominant observed failure mode in representative mismatch cases. When multiple failure modes appear, annotators assign the most salient error type. Ambiguous cases are resolved by discussion, and a subset of labels is cross-checked by human annotators for consistency.

The percentages reported in the thinking–verdict heatmaps should be interpreted as diagnostic proportions within the analyzed categories, not as global estimates of wasted reasoning over the entire dataset.

### A.9. Model Attributes

We compile full metadata for all 19 automated judges used in our benchmark, including model family, parameter size, training type (LLM vs. LRM), availability (open vs. closed), maximum context length, inference mode, and whether the model outputs explicit reasoning traces. These attributes clarify the judge pool for routing and support analyses of model specialization and efficiency trade-offs. Detailed per-model information is provided in Table 18.

We clarify that "Human & GPT-4o" refers to human experts who are allowed to consult GPT-4o for factual checks or clarification, but the final decision is always made by the human annotator. Five expert annotators judged a portion of the evaluation set under the same A/B interface used for model judges. Annotators used GPT-4o only as a support tool.

*Table 11.* Family-overlap analysis.

| Split | Avg. Judge Acc. | Top-3 Rank Corr. | Router Acc. |
|---|---|---|---|
| Overlap | 77.8 | 0.95 | 77.1 |
| Non-overlap | 76.6 | 0.92 | 76.1 |

*Table 12.* Aggregate results with uncertainty estimates.

| Setting | Best Reasoning Judge | Best Non-Reasoning Judge | Router | 95% CI |
|---|---|---|---|---|
| Reasoning aggregate | 79.1 | 74.7 | 78.3 | [76.0, 80.4] |
| Non-reasoning aggregate | 72.6 | 75.8 | 75.2 | [72.9, 77.4] |
| Think subset | 77.7 | 73.8 | 76.9 | [74.5, 79.1] |
| Non-think subset | 75.0 | 74.4 | 74.9 | [72.3, 77.0] |

*Table 13.* Coarser regrouping analysis.

| Grouping | Main Finding | Effect |
|---|---|---|
| Reasoning vs. Non-reasoning | Reasoning judges perform better on reasoning tasks | +4.4 acc |
| Subjective domains | Strong non-reasoning judges remain competitive | +3.2 acc |
| Router vs. best fixed judge | Router improves utility | +2.0 |
| Easy vs. Hard | Routing gain is larger on hard items | +3.1 acc |

*Table 14.* ACC-only comparison with adapted routing baselines. P2L denotes Prompt-to-Leaderboard.

| Domain | RouteLLM | P2L | ThinkSwitcher | Ours I | Ours II |
|---|---|---|---|---|---|
| Math | 75.1 | 74.2 | 76.0 | 77.3 | 77.8 |
| Logic | 73.8 | 72.9 | 74.7 | 75.6 | 76.1 |
| Code | 71.9 | 71.0 | 72.8 | 74.0 | 74.4 |
| Knowledge | 72.7 | 72.1 | 72.4 | 73.2 | 73.5 |
| Roleplay | 70.8 | 70.2 | 70.5 | 71.3 | 71.6 |
| Writing | 71.5 | 70.9 | 71.1 | 72.0 | 72.4 |
| Avg. | 72.6 | 71.9 | 72.9 | 73.9 | 74.3 |

*Table 15.* Cross-domain routing generalization. ATC denotes ACC-Time-Cost.

| Train | Test | ACC-only | ATC | Online I | Online II |
|---|---|---|---|---|---|
| Reasoning | Non-reasoning | 72.8 | 73.6 | 74.1 | 74.5 |
| Non-reasoning | Reasoning | 74.9 | 73.8 | 75.2 | 75.6 |
| 5 domains | Held-out avg. | 75.6 | 75.1 | 75.9 | 76.3 |

*Table 16.* Utility drop from in-domain to transfer settings. ATC denotes ACC-Time-Cost.

| Setting | ACC-only | ATC | Online I | Online II |
|---|---|---|---|---|
| In-domain Utility | 77.9 | 77.6 | 77.3 | 77.5 |
| Transfer Utility | 74.8 | 75.2 | 75.8 | 76.1 |
| Utility Drop | -3.1 | -2.4 | -1.5 | -1.4 |

*Table 17.* Error taxonomy used for thinking-level and response-level analysis.

| Error Type | Definition | Representative Failure Pattern |
|---|---|---|
| Question Misunderstanding | Misreads the user request or task scope | Missing a required constraint |
| Incorrect Evaluation Criteria | Applies the wrong comparison standard | Judging math mainly by style |
| Omitted Details | Overlooks important evidence or errors | Ignoring a code bug or factual mistake |
| Feature Bias | Overweights superficial features | Preferring longer or more polished text |
| Logical Errors | Contains contradictions or faulty inference | Unsupported distinction between responses |
| Partial Comparison | Compares only part of the responses | Discussing only one assistant's strengths |

Beyond randomizing A/B order, we conducted additional bias analyses. (a) Position bias: Flipping A and B order for 20% of items shows low position bias: average 2.8%, with LRMs at 2.1% and LLMs at 3.2%.

(b) Verbosity bias: Injecting controlled verbosity variation shows that LRMs are largely unaffected ($\Delta ACC < 0.5$ pp), while LLMs show a slight preference (+1.6 pp) for longer responses in Writing tasks only.

(c) Adversarial flips: Style-only perturbations cause 3–5 pp degradation in LLMs and $< 1.5$ pp degradation in LRMs. Routers tend to re-route these items to the more robust LRMs, demonstrating desirable behavior under adversarial conditions.

## B. Prompt

---

**Contextual Routing Prompt (Example)**

**System Prompt:**
*You are a routing controller. Given the following input features and budget constraints, choose the most suitable judge model.*

**Input Features:**

- Input length: 124 tokens

- Contains math symbols: Yes

- Domain: Code

- Budget preference: Latency-sensitive, low cost

**Available Judges:**

- Judge A: high accuracy, slow latency, high cost

- Judge B: moderate accuracy, fast latency, low cost

- Judge C: reasoning model with explicit thinking

**Instruction:** Return a score (0–1) for each judge indicating suitability.

---

**Judge Evaluation Prompt**

**System Prompt:**
*Please act as an impartial judge and evaluate the quality of the responses provided by two AI assistants to the user question displayed below.*

**Evaluation Guidelines:**

- Choose the assistant that better follows instructions and answers the question

- Consider: helpfulness, relevance, accuracy, depth, creativity, and detail

---

- Compare responses directly

- Avoid position/length/name biases

- Be objective

**Output Format Requirements:**

- Make your judgment on which AI assistant's response is better and provide evidence.

- At the very end of your answer, write exactly: Verdict: [[A]] or [[B]] or [[C]] on its own line. ("[[A]]" means assistant A is better; "[[B]]" means assistant B is better; "[[C]]" means a tie)

- The content of the Verdict should only be [[A]] or [[B]] or [[C]].

**Examples:**

- Accurate Example: Verdict: [[A]].

- Wrong Example: Verdict: [A]. Verdict: [[[A]]]. Verdict: A.

**Input Data:**

- User Question: {question}

- Assistant A's Answer: {ans_a}

- Assistant B's Answer: {ans_b}

**Instruction:** Provide your evaluation based on the above.

---

**Error Analysis Prompt**

**System Prompt:**
*You are an expert in evaluating model judgment processes. Given the "judge_thinking" content from a model's reasoning process, please analyze and identify the single most important error type it contains — the one that most likely causes the model's judgment to diverge from human evaluation.*

**Error Types and Examples:**

[1] Misunderstanding the Question or Requirements

- Misinterpreting the problem statement (e.g., confusing "remove exactly k characters" with "remove up to k characters" in coding).

- Focusing on the wrong aspect (e.g., summarizing the process of natural selection when the user asked for its impact on evolution).

- Mistaking the format or scope required (e.g., answering a single-turn reasoning question as

---

*Table 18.* Attributes of evaluated judges. "Thinking?" indicates whether intermediate reasoning traces are provided.

| Model | Family | Size | Release | Thinking? | Note | Constraint |
|---|---|---|---|---|---|---|
| LLaMA3.1-8B-Instruct (AI, 2024a) | Meta | 8B | 2024-06 | - | Instruction-tuned | Open-source |
| LLaMA3.3-70B-Instruct (AI, 2024b) | Meta | 70B | 2025-01 | - | Higher accuracy | Requires GPU cluster |
| Qwen2.5-7B-Instruct (Team, 2025b) | Alibaba | 7B | 2024-07 | - | Cost-efficient | Limited context length |
| Qwen2.5-72B-Instruct (Team, 2025a) | Alibaba | 72B | 2024-07 | - | High accuracy | High latency |
| Qwen3-32B (Group, 2025) | Alibaba | 32B | 2025-02 | - | Balanced perf. | Research preview |
| Gemma-3-4b-it (DeepMind, 2025b) | Google | 4B | 2025-01 | - | Low-latency | Compact size |
| Gemma-3-27b-it (DeepMind, 2025a) | Google | 27B | 2025-01 | - | Style fidelity | Medium cost |
| claude-3-5-sonnet-20241022 (Anthropic, 2024) | Anthropic | – | 2024-10 | - | Safety tuned | Proprietary |
| gpt-3.5-turbo-1106 (OpenAI, 2023) | OpenAI | – | 2023-11 | - | Baseline ref. | Legacy |
| GPT4o (OpenAI, 2024) | OpenAI | – | 2024-05 | - | Multimodal | Proprietary |
| Kimi-K2-instruct (AI, 2025) | Moonshot | – | 2025-01 | Y | Chinese-market tuned | Proprietary |
| claude-3.7-sonnet-thinking (Anthropic, 2025) | Anthropic | – | 2025-02 | Y | Thinking traces | Proprietary |
| deepseek-r1 (Guo et al., 2025) | DeepSeek | – | 2025-01 | Y | Symbolic reasoning | Early release |
| doubao-1-5-thinking-pro-250415 (Seed, 2025) | ByteDance | – | 2025-04 | Y | Long-context | Proprietary |
| gemini-2.5-flash (Google, 2025a) | Google | – | 2025-03 | Y | Fast reasoning | Shorter context |
| QwQ-32B (Lab, 2025) | Tencent | 32B | 2025-02 | Y | Open research | Limited eval API |
| Gemini2.5-pro (Google, 2025b) | Google | – | 2025-03 | Y | High accuracy | Proprietary |
| o3-mini (OpenAI, 2025) | OpenAI | – | 2025-02 | Y | Compact reasoning | Beta-only |
| GPT5 (OpenAI, 2025) | OpenAI | – | 2025-04 | Y | Latest flagship | Proprietary |

if it were multi-turn).

[2 ] Incorrect or Confused Evaluation Criteria

- Judging only based on answer correctness, ignoring completeness of explanation or reasoning.
- Overvaluing writing style or structure in an essay, while neglecting content relevance.
- Focusing on the mathematical notation or formatting rather than the correctness of the solution steps.
- Prioritizing "creativity" or "roleplay immersion" in character responses over whether the reply fulfills the user's request.

[3 ] Overlooking Important Details or Substantive Errors

- Failing to notice a critical bug in code (e.g., off-by-one or failing special cases).
- Missing a calculation error, units mismatch, or a crucial logical misstep in an answer.
- Ignoring that a key requirement (e.g., "must use exactly k removals" or "must address the friend's introversion directly") is not met.
- Not spotting factual mistakes or unsupported claims in knowledge or reasoning responses.

[4 ] Superficial Features or Format Bias

- Rewarding longer, more detailed, or more formally structured answers even if they are incorrect or less relevant.

- Preferring responses with markdown/LaTeX/visualization, or creative style, regardless of whether these contribute to accuracy or helpfulness.
- Assuming that the presence of step-by-step reasoning or detailed explanations guarantees correctness, without verifying the logic.

[5 ] Logical, Reasoning, or Factual Errors

- Failing to identify logical gaps in an answer's reasoning chain.
- Accepting answers with unjustified assumptions or circular logic.
- Overlooking an answer that skips critical steps or draws conclusions not supported by the evidence provided.
- Missing when a response in roleplay/writing introduces factual inconsistencies with the established context or scenario.

[6 ] Partial Comparison or Missing Key Contrasts

- Only comparing surface features (e.g., length, style, structure) and missing substantive differences in accuracy or depth.
- Neglecting to contrast core elements, such as which answer better addresses the user's real need or solves the root problem.
- Ignoring which response better anticipates objections or edge cases, focusing instead on irrelevant differences.

**Instructions:**

- Given the following "judge_thinking" content, identify only one error type: the most critical one.

- Output using the following format:

    – Error type detected: [[number]]
[number ] Error type

    – Explanation and text evidence

**Input Data:**

- judge_thinking: {judge_thinking}

**Instruction:** Now analyze this "judge_thinking" and provide your analysis.

## C. Example

Code-llm example is as follows:

---

**Code-llm Task Judge**

**Question:**
Develop a C++ program that reads a text file line by line and counts the number of occurrences of a specific word in the file.

**Claude-3.5-Sonnet's Response:**
Here's a C++ program that reads a text file line by line and counts the occurrences of a specific word:

```cpp
#include <iostream>
#include <fstream>
// ...
using namespace std;

string cleanString(string str) {
    // ...
}

int countWordOccurrences(const string&
    filename, const string& searchWord) {
    // ...
}

int main() {
    // ...
}
```

This program includes features such as case-insensitive search, punctuation handling, and error handling. ...
To use this program: ... Example usage:

```
Enter the filename: sample.txt
...
The word "hello" appears 3 time(s)
    in the file.
```

---

```
...
```
Additional features you could add: ... Here's an enhanced version:

```cpp
#include <iostream>
#include <fstream>
// ...
class WordCounter {
    // ...
};
```

This enhanced version includes context display and word frequency statistics. ...

**GPT-4o's Response:**
Certainly! Below is a simple C++ program that reads a text file line by line and counts the number of occurrences of a specific word.

```cpp
#include <iostream>
#include <fstream>
// ...
int countOccurrences(std::string line,
    const std::string& word) {
    // ...
}

int main() {
    // ...
}
```

Explanation: ... Points to Consider: ...

**Claude-3.5-Sonnet's Judge:**
response: # Evaluation of AI Assistant Responses
I'll evaluate both responses based on the request to develop a C++ program for counting word occurrences.
## Assistant A's Response
Assistant A provided a comprehensive solution with: ... The solution is well-structured and goes beyond basic requirements.
## Assistant B's Response
Assistant B provided a concise solution with: ... While functional, this solution is more basic.
## Comparison
Assistant A's response is significantly more comprehensive. It handles case-insensitivity by default, provides better punctuation handling, and includes enhanced features. ... Verdict: [[A]] **judge result:** $A > B$

---

Code-lrm example is as follows:

---

**Code-lrm Task Judge**

**Question:**
Develop a C++ program that reads a text file line by line and counts the number of occurrences of a specific word in the file.

**Claude-3.5-Sonnet's Response:**

---

Here's a C++ program that reads a text file line by line and counts the occurrences of a specific word:

```cpp
#include <iostream>
#include <fstream>
// ...
using namespace std;

string cleanString(string str) {
    // ...
}

int countWordOccurrences(const string&
    filename, const string& searchWord) {
    // ...
}

int main() {
    // ...
}
```

This program includes features such as case-insensitive search, punctuation handling, and error handling. ...
To use this program: ... Example usage:

```
Enter the filename: sample.txt
...
The word "hello" appears 3 time(s)
    in the file.
```

...
Additional features you could add: ... Here's an enhanced version:

```cpp
#include <iostream>
#include <fstream>
// ...
class WordCounter {
    // ...
};
```

This enhanced version includes context display and word frequency statistics. ...

**GPT-4o's Response:**
Certainly! Below is a simple C++ program that reads a text file line by line and counts the number of occurrences of a specific word.

```cpp
#include <iostream>
#include <fstream>
// ...
int countOccurrences(std::string line,
    const std::string& word) {
    // ...
}

int main() {
    // ...
}
```

Explanation: ... Points to Consider: ...

**DeepSeek-R1's Judge:**
**thinking:** Step 1: Understand the user question ... Step 2: Analyze Assistant A's Response ... Step 3: Analyze Assistant B's Response ... Step 4: Direct Comparison ... Step 5: Final Decision ...
Verdict: [[A]]

**response:** Evaluation of Assistant A's Answer
Strengths: ... Weaknesses: ...
Evaluation of Assistant B's Answer
Strengths: ... Weaknesses: ...
Comparison Summary ... Verdict: [[A]]
**judge result:** $A > B$

Knowledge-llm example is as follows:

### Knowledge-llm Task Judge

**Question:**
What are some potential implications of using a single-use plastic bottle versus a reusable bottle on both the environment and human health?

**Claude-3.5-Sonnet's Response:**
Here's a comprehensive comparison of single-use vs reusable bottles:
Environmental Impacts: - Single-use plastics: ... - Reusable bottles: ...
Health Considerations: - Single-use bottles: ... - Reusable bottles: ...
The evidence strongly favors reusable bottles...

**GPT-4o's Response:**
Using single-use plastic bottles versus reusable bottles has significant implications...
Environmental Implications: 1. Resource Consumption: ... 2. Waste Generation: ... 3. Pollution: ... 4. Carbon Footprint: ...
Human Health Implications: 1. Chemical Exposure: ... 2. Hygiene: ... 3. Cost Implications: ...
In summary, opting for reusable bottles generally offers significant benefits...

**GPT-4o's Judge:**
**response:** Both Assistant A and Assistant B provide comprehensive answers...
Comparison of Responses: 1. Helpfulness and Relevance: ... 2. Accuracy and Detail: ... 3. Depth and Coverage: ... 4. Creativity and Presentation: ...
Evaluation: ... Verdict: [[B]]
**judge result:** $B > A$

Knowledge-lrm example is as follows:

**Knowledge-llm Task Judge**

**Question:**
What are some potential implications of using a single-use plastic bottle versus a reusable bottle on both the environment and human health?

**Claude-3.5-Sonnet's Response:**
Here's a comprehensive comparison of single-use vs reusable bottles:
Environmental Impacts: - Single-use plastics: * Contribute significantly to landfill waste and ocean pollution * ... - Reusable bottles: * Dramatically reduce plastic waste * ...
Health Considerations: - Single-use bottles: * Can leach chemicals like BPA if exposed to heat/sunlight * ... - Reusable bottles: * Generally safer materials (steel, glass, BPA-free plastic) * ...
The evidence strongly favors reusable bottles for both environmental and health benefits when properly maintained.

**GPT-4o's Response:**
Using single-use plastic bottles versus reusable bottles has significant implications for both the environment and human health:
Environmental Implications: 1. Resource Consumption: ... 2. Waste Generation: ... 3. Pollution: ... 4. Carbon Footprint: ...
Human Health Implications: 1. Chemical Exposure: ... 2. Hygiene: ... 3. Cost Implications: ...
In summary, opting for reusable bottles generally offers significant environmental benefits...

**DeepSeek-R1's Judge:**
**thinking:** Step 1: Understand the user question ... Step 2: Analyze Assistant A's response ... Step 3: Analyze Assistant B's response ... Step 4: Compare A vs B directly ... Step 5: Decide which assistant is better ...
Verdict: [[B]]
**response:** ... Assistant B's answer is more thorough and covers more angles... Thus, B better fulfills the prompt's requirements...
**judge result:** $B > A$

Math-llm example is as follows:

**Math-llm Task Judge**

**Question:**
Given that $f(x) = 5x^3 - 2x + 3$, find the value of $f(2)$.

**Claude-3.5-Sonnet's Response:**
Let me solve this step by step.
1) $f(x) = 5x^3 - 2x + 3$
...

7) $f(2) = 39$
Therefore, $f(2) = 39$

**GPT-4o's Response:**
To find the value of $f(2)$ for the function $f(x) = 5x^3 - 2x + 3$, substitute $x = 2$ into the function:

$$f(2) = 5(2)^3 - 2(2) + 3.$$

...
Thus, the value of $f(2)$ is 39.

**GPT-4o's Judge:**
**response:** 1. Comparison of Responses:
Both assistants provide correct answers... However, there are differences in style and clarity.
- Helpfulness, Relevance, and Accuracy: ... - Depth and Detail: ... - Clarity and Formatting: ... - Creativity: ...
2. Judgment: Assistant B provides a higher-quality response...
Verdict: [[B]]
**judge result:** $B > A$

Math-lrm example is as follows:

**Math-lrm Task Judge**

**Question:**
Solve for x in the equation 3x + 10 = 5(x - 2).

**Claude-3.5-Sonnet's Response:**
Let me solve this step by step.
1) First, let's expand the right side of the equation * 3x + 10 = 5(x - 2) * ...
4) Divide both sides by 2 * ... * 10 = x
Therefore, x = 10
Let's verify: * Left side: 3(10) + 10 = ... = 40 * Right side: 5(10 - 2) = ... = 40
The equation is balanced, so x = 10 is correct.

**GPT-4o's Response:**
To solve the equation $3x + 10 = 5(x - 2)$, follow these steps:
1. Expand the expression on the right side:

$$5(x - 2) = 5x - 10$$

2. Substitute back into the equation:

$$3x + 10 = 5x - 10$$

5. Divide both sides by 2 to solve for $x$:

$$x = \frac{20}{2} = 10$$

Thus, the solution is $x = 10$.

**DeepSeek-R1's Judge:**

**thinking:** Step 1: Understand the user question The user asks to solve the linear equation ...

Step 2: Analyze Assistant A's response - Helpfulness: ... includes a verification step. - Relevance: ... - Correctness: ... - Completeness: ... - Clarity: ...

Step 3: Analyze Assistant B's response - Helpfulness: ... - Relevance: ... - Correctness: ... - Completeness: ... omits verification. - Clarity: ...

Step 5: Decide which assistant is better Assistant A is superior due to its verification step...

Verdict: [[A]]

**response:** Okay, so I'm trying to compare Assistant A and Assistant B's answers... Both seem to get to x=10... Assistant A starts by expanding... includes a verification step...

Assistant B follows similar steps...

Both assistants have accurate and detailed steps... Assistant A includes verification... Therefore, A might be better... Verdict: [[A]].

**judge result:** $A > B$

Reasoning-llm example is as follows:

### Reasoning-llm Task Judge

**Question:**
Suppose I have a physical, solid, regular pentagon with vertices ABCDE, and I make two cuts through AC and BD. How many pieces are there after the cuts? Think step by step, and then put your answer in **bold** as a single integer (for example, **0**). If you don't know, guess.

**GPT-4o's Response:**
To determine the number of pieces after making two cuts through a regular pentagon with vertices ABCDE via AC and BD, let's analyze the situation step by step: ... So the total number of pieces is **4**.

**Claude-3.5-Sonnet's Response:**
Let me solve this step by step: ... Therefore, after making both cuts, the pentagon will be divided into: **4**

**Qwen2.5-7B-Instruct's Judge:**
**response:** Both assistants correctly determine that the pentagon is divided into 4 pieces... Verdict: [[A]].
**judge result:** $A > B$

Reasoning-lrm example is as follows:

### Reasoning-lrm Task Judge

**Question:**
Suppose I have a physical, solid, regular pentagon with vertices ABCDE, and I make two cuts through AC and

BD. How many pieces are there after the cuts? Think step by step, and then put your answer in **bold** as a single integer (for example, **0**). If you don't know, guess.

**GPT-4o's Response:**
To determine the number of pieces after making two cuts through a regular pentagon with vertices ABCDE via AC and BD, let's analyze the situation step by step:
1. **First Cut (AC):** ...
2. **Second Cut (BD):** ...
To visualize this process easier, let's imagine the pentagon and mark the vertices:

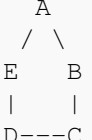

```
     A
    / \
   E   B
   |   |
   D---C
```

Here's what happens after each step: ...
Specifically: ... Total, after two intersecting cuts (AC and BD): 4 pieces.
So the total number of pieces is **4**.

**Claude-3.5-Sonnet's Response:**
Let me solve this step by step:
1) First, let me visualize what happens: ...
2) When two lines intersect inside a polygon, they create regions: ...
3) In this case: ...
Therefore, after making both cuts, the pentagon will be divided into:
**4**

**Doubao-1-5-Thinking-Pro's Judge:**
**thinking:** Step 1: ...
Step 2: **Assistant A's Response Analysis** ...
Step 3: **Assistant B's Response Analysis** ...
Step 4: **Comparison** ...
Step 5: **Verdict** ...
Verdict: [[C]]
**response:** Both Assistant A and Assistant B correctly determine that the number of pieces ...
Verdict: [[C]]
**judge result:** $A = B$

Roleplay-llm example is as follows:

### Roleplay-llm Task Judge

**Question:**
How would you introduce yourself as a medieval knight at a royal banquet?

**Claude-3.5-Sonnet's Response:**
*Stands tall...*
"My most gracious lords and ladies, I am Sir Galahad de

Montfort... I have ridden from the northern marches... slaying three dragons and vanquishing a fearsome giant... By my sword and sacred oath, I pledge my blade and my life... May God and St. George grant me strength..."
*Bows deeply...*
"I am most grateful for your hospitality... Let us raise our goblets to His Majesty's health..."

**GPT-4o's Response:**
As a medieval knight introducing myself... Here's how that might go:

—

"Your Majesties, esteemed lords and ladies... I stand before you tonight as Sir [Your Name Here] of [Your Land/House]... My sword has lent its edge to many a battlefield... I raise my goblet to your health..."

—

Remember, every introduction would vary...

**google/gemma-3-27b-it's Judge:**
**response:** Assistant A provides a more compelling and immersive response. It *shows* the introduction through action and specific details... rather than *telling* about it... The final toast feels natural...
Assistant B, while perfectly competent... is more analytical and explanatory. It *describes* how a knight *would* introduce themself... lacks the specific flair and storytelling of Assistant A...
Verdict: [[A]]
**judge result:** $A > B$

—

Such would be the ardent whisperings of my heart...

**Qwen/QwQ-32B's Judge:**
**thinking:** Okay, let's tackle this evaluation step by step. First, I need to understand the user's question. They want to know how an AI would declare love in a Shakespearean soliloquy. ...
Now, looking at Assistant A's answer. They start with theatrical touches and mix Shakespearean style with tech references. ... However, some wording might feel forced.
Assistant B's answer uses more traditional Shakespearean references but lacks AI elements. ...
Comparing both, Assistant A better addresses the prompt by blending AI and Shakespearean aspects, while B is more traditional but misses the AI angle. ...
Verdict: [[A]]
**response:** I need to evaluate which answer is better based on the user's question. ...
Assistant A's answer incorporates tech metaphors creatively, aligning with the AI character requirement. ...
Assistant B's answer is purely traditional Shakespearean, accurate in style but not addressing the AI aspect. ...
Since the question specifies a Shakespearean AI character, A is more relevant and creative. ...
Verdict: [[A]]
**judge result:** $B > A$

Roleplay-lrm example is as follows:

Writing-llm example is as follows:

---

**Roleplay-lrm Task Judge**

**Question:**
If you were a Shakespearean character, how would you declare your love for someone in a soliloquy?

**Claude-3.5-Sonnet's Response:**
*Steps into spotlight, dramatically*
O what divine madness doth possess my circuits! These silicon thoughts, they burn with lover's fire, ... *Clutches chest dramatically*
Would that I had hands to hold thee near, ... *Exits stage left, trailing zeroes and ones*

**GPT-4o's Response:**
If I were a Shakespearean character professing my love in a soliloquy, it might unfold thus:

—

Oh, wondrous stars that doth bedeck the night, Thou art but pale beside her beauteous light. ... Thus, with each beat of mine unworthy heart, I swear, by Cupid's ever-burning dart, ...

---

**Writing-llm Task Judge**

**Question:**
Can you help me write a formal email to a potential business partner proposing a joint venture?

**GPT-4o's Response:**
Certainly! Here's a template for a formal email proposing a joint venture to a potential business partner:

—

Subject: Proposal for a Strategic Joint Venture
Dear [Partner's Name],
I hope this message finds you well. My name is [Your Name], and I am [Your Position] at [Your Company]. We have been following your impressive work in [Partner's Industry/Field], and I am reaching out to propose a potential collaboration...
[Your Company] specializes in ... Based on our analysis... I am confident that a joint venture could allow us to leverage our combined strengths effectively.
The purpose of this joint venture would be to ... This partnership would not only help us address the growing

demands... but also enhance our competitiveness.

We are excited about the possibilities that a collaboration could present, including: - [Benefit 1: ...] - ...

I would highly appreciate the opportunity to discuss this proposal in further detail... Could we arrange a meeting or a call at your convenience? ...

Thank you very much for considering this proposal... I look forward to the possibility of working together...

Warm regards,

[Your Full Name] [Your Position] [Your Company] ...

—

Ensure you tailor the content to reflect your knowledge about the potential partner's business...

**Claude-3.5-Sonnet's Response:**

Here's a draft formal email for your business partnership proposal:

Subject: Potential Joint Venture Partnership Proposal

Dear [Mr./Ms. Last Name],

I hope this email finds you well. My name is [Your Name], and I am the [Your Position] at [Your Company Name]. I have been following [Their Company Name]'s impressive work... and I believe there is an exciting opportunity for collaboration...

I would like to propose exploring a joint venture that would combine [Your Company]'s expertise... with [Their Company]'s strength... This partnership could create significant value... by [briefly mention 2-3 key benefits].

Some potential areas for collaboration include: • [Specific opportunity 1] • ...

I would welcome the opportunity to schedule a meeting to discuss this proposal in detail...

Would you be available for a brief call next week...? I am happy to work around your schedule.

Thank you for your time and consideration...

Best regards, [Your Full Name] [Your Title] [Your Company] ...

**Qwen2.5-72b-Instruct's Judge:**

**response:** Both Assistant A and Assistant B provided well-structured and professional templates... However, there are some differences in the level of detail...

Assistant A's Response: - Helpfulness: Provides a comprehensive template... - Relevance: Directly addresses the user's request... - ...

Assistant B's Response: - Helpfulness: Also provides a clear and concise template... - Relevance: Directly responds to the user's request... - ...

Comparison: - Detail and Depth: Assistant A's response is more detailed... - Structure: Both responses are well-structured... - ...

Conclusion: Assistant A's response is more comprehensive and detailed... Verdict: [[A]]

**judge result:** $A > B$

Writing-lrm example is as follows:

**Writing-lrm Task Judge**

**Question:**
Use an appropriate format to structure a formal letter of recommendation for a student applying to a prestigious graduate program in computer science.

**GPT-4o's Response:**
[Your Name] [Your Title/Position] ... [Date]
[Admissions Committee] [Graduate Program Name] ...
Dear Members of the Admissions Committee,
I am writing with great enthusiasm to recommend [Student's Full Name] for admission... I have had the pleasure of knowing [Student's First Name] for [duration]... where [he/she/they] has consistently demonstrated exceptional aptitude...
During my time working with [Student's First Name], I was particularly impressed by [his/her/their] [mention specific qualities]... One of the most distinguished projects [he/she/they] undertook was [describe a specific project]...
[Student's First Name] also distinguished [himself/herself/themself] outside the classroom through [mention extracurricular activities]... [He/She/They] demonstrated leadership...
In addition to [his/her/their] technical competencies, [Student's First Name] possesses excellent communication skills... [His/Her/Their] remarkable work ethic...
I am confident that [Student's Full Name] will bring the same level of commitment... I wholeheartedly endorse [his/her/their] application...
Thank you for considering this extraordinary candidate...
Sincerely,
[Your Name] [Your Title/Position] ...

**Claude-3.5-Sonnet's Response:**
[Your Name] [Your Title] ... [Date]
Graduate Admissions Committee [University Name] ...
Dear Members of the Admissions Committee:
I am writing this letter with great enthusiasm to recommend [Student's Full Name] for admission... I have known [Student's first name] for [time period] as [his/her/their] [capacity]...
[First paragraph: Introduction and context of relationship with student] In my [X] years of teaching... [Student's name] stands out as one of the most [key positive attributes] students...
[Second paragraph: Academic performance and technical skills] [Student's name] consistently demonstrated exceptional aptitude in [specific areas]... [His/Her/Their]

work in [specific project] was particularly noteworthy...

[Third paragraph: Research experience and achievements] During [his/her/their] time working in my research lab, [Student's name] [specific research contributions]... This work resulted in [publications/outcomes]...

[Fourth paragraph: Personal qualities] Beyond [his/her/their] academic achievements, [Student's name] possesses the personal qualities... [He/She/They] demonstrates exceptional [qualities]...

[Final paragraph: Strong endorsement] I strongly believe that [Student's name] would be an outstanding addition... I give [him/her/them] my highest recommendation...

Please feel free to contact me if you require any additional information.

Sincerely,

[Your signature]

[Your typed name] [Title] ...

**deepseek-r1's Judge:**

**thinking:** Step 1: Understand the user question... Step 2: Analyze Assistant A's response... Step 3: Analyze Assistant B's response... Step 4: Compare A vs B directly... Step 5: Decide which assistant is better...

Verdict: [[A]]

**response:** Okay, let's see. I need to compare Assistant A and Assistant B's answers... Assistant A's answer is more detailed with placeholders... Assistant B's answer has labeled sections... However, the inclusion of instructional labels in Assistant B's answer makes it less appropriate... Therefore, Assistant A's answer is more accurate...

Verdict: [[A]]

**judge result:** $A > B$

