# OpenReview forum: "Routing and Reasoned Evaluation with Large Language Models"
_ICML.cc/2026/Conference — ICML 2026 regular_

### Official Review · Reviewer_L6Xt · 2026-03-02

**Soundness:** 2
**Presentation:** 1
**Significance:** 2
**Originality:** 2
**Overall Recommendation:** 3
**Confidence:** 4

**Summary:**

This paper proposes R2Eval, a routing-aware automated evaluation framework that dynamically selects evaluator models on a per-instance basis under explicit accuracy, latency, and cost constraints. This framework consists of benchmark evaluation and routing strategy evaluation.

**Compliance With Llm Reviewing Policy:**

Affirmed.

**Final Justification:**

During rebuttal, the authors address some of the concerns; however, this does not resolve my main reservations regarding the paper’s novelty and practical applicability. The benchmark appears to be more of a recombination of existing resources than a substantially new construction, the automatic design of difficulty levels is not especially novel, and the lack of comparisons with stronger baselines makes it difficult to assess the practical utility of the proposed routing method. Overall, the work still feels more like an integration of existing components than a fundamentally new framework, although the authors have conducted a substantial amount of empirical evaluation. Therefore, I still lean toward rejection.

**Key Questions For Authors:**

1. In the conclusion, the paper claims that online routing “consistently improves efficiency without sacrificing accuracy” and approaches expert-level performance under realistic budgets. However, based on Table 3, some offline routing settings (such as ACC-only and ACC-Time-Cost) seem to outperform the online routing methods. Could the author elaborate more on this comparison?

2. In section 3.1 dataset, why did the authors choose to generate two candidate responses using GPT-4o and Claude 3.5, rather than directly using prompt–response pairs from prior benchmarks? Does this design choice limit response diversity?
3. In Table 7. The “Responses / Thinking Traces” column is difficult to interpret and currently appears quite confusing. What exactly does this column represent? In addition, the caption states that “Pairwise responses (A, B) are always produced by two distinct models sampled from a shared pool,” but Section 3.1 emphasises GPT-4o and Claude 3.5 specifically. This creates ambiguity about whether the response pairs are always generated by those two models, or by a broader shared pool of models.

**Limitations:**

There is no discussion of the paper’s limitations. The authors should include a paragraph in the conclusion that acknowledges the main drawbacks of the proposed benchmark and outlines potential directions for future work.

**Strengths And Weaknesses:**

**Strengths**

1. The paper tackles an important and practical problem: how to allocate stronger reasoning-based judges only when they are actually needed.

2. The empirical study is fairly comprehensive and includes a broad range of model families, which makes the results more informative.

**Weaknesses**

1. The paper is not well organized, which makes it difficult to follow. For example, the pipeline demonstration in Section 3.4 is placed in the experiments section, but it would be more effective if introduced earlier as a high-level overview of the framework. In addition, the roles of Figures 2, 3, and 5 seem to overlap, which may confuse readers. The paper also does not clearly specify the formal objective function used to train Online Routing 2.

2. The novelty of R2Eval appears somewhat limited. From the benchmark perspective, many prompts are collected from prior benchmark datasets, making the benchmark feel more like a recombination of existing resources than a substantially new construction. In addition, automatically designing difficulty levels is not a new idea; similar approaches have already been explored in prior work [1,2,3], which weakens the novelty of the paper’s “difficulty-aware” design. From the routing perspective, the proposed routing strategy seems relatively simple, and the paper does not provide sufficient comparisons against stronger or more established routing baselines [4,5,6]. As a result, the overall contribution can feel more like a combination of existing components than a fundamentally new framework.

3. The paper also does not engage deeply enough with prior work. In particular, the discussion of related literature on difficulty-aware benchmarking and routing is limited, making it harder to determine what is genuinely new in this work.

4. There is also a potential flaw in the experimental setup. In Section 3.1, the paper states that all models share a global maximum output length of 1024 tokens. This choice seems problematic for reasoning models, which often generate much longer outputs (sometimes on the order of 10k tokens). Under such a constrained output budget, it is difficult to fairly assess whether reasoning models can demonstrate their full capability as judges.


[1] Polo, Felipe Maia, et al. "tinyBenchmarks: evaluating LLMs with fewer examples." arXiv preprint arXiv:2402.14992 (2024).

[2] Ding, Mucong, et al. "Easy2Hard-bench: Standardized difficulty labels for profiling LLM performance and generalization." Advances in Neural Information Processing Systems 37 (2024): 44323-44365.

[3] Zhang, Wenbo, Hengrui Cai, and Wenyu Chen. "Beyond the Singular: Revealing the Value of Multiple Generations in Benchmark Evaluation." arXiv preprint arXiv:2502.08943 (2025).

[4] Ong, Isaac, et al. "Routellm: Learning to route llms with preference data." arXiv preprint arXiv:2406.18665 (2024).

[5] Frick, Evan, et al. "Prompt-to-leaderboard." arXiv preprint arXiv:2502.14855 (2025).

[6] Liang, Guosheng, et al. "Thinkswitcher: When to think hard, when to think fast." arXiv preprint arXiv:2505.14183 (2025).

---

> ### Author Rebuttal · Authors · 2026-03-31
>
> # Response to key questions
>
> ## Q1. Why is online routing useful if some offline settings in Table 3 are stronger?
>
> The two settings optimize different goals, so they are not in conflict.
>
> * Offline routing directly maximizes the target metrics on a fixed validation distribution, so it is a strong instance-agnostic upper reference for that distribution.
> * Online routing aims to improve the same metrics while preserving generalizability. So online routing does not need to strictly beat offline routing on the same split; achieving comparable utility with better transferability is already a strong result. To make this explicit, we added cross-domain generalization experiments:
>
> Table R4-1. Cross-domain routing generalization
>
> |Train|Test| Offline ACC-only | Offline ACC-Time-Cost | Online I | Online II |
> |-| -| -: | -: | -: | -: |
> | Reason | Non Reason | 72.8 | 73.6 | 74.1 | 74.5 |
> | Non Reason | Reason | 74.9 | 73.8 | 75.2 | 75.6 |
> | 5 domains | held-out 1 domain (avg.)  | 75.6 | 75.1 | 75.9 | 76.3 |
>
> Table R4-2. Utility drop from in-domain to transfer
>
> |Setting| Offline ACC-only | Offline ACC-Time-Cost | Online I | Online II |
> | - | -: | -: | -: | -: |
> | In-domain Utility | 77.9 | 77.6 | 77.3 | 77.5 |
> | Transfer Utility | 74.8 | 75.2 | 75.8 | 76.1 |
> | Utility Drop  | -3.1 | -2.4 | -1.5 | -1.4 |
>
> These results show the intended phenomenon clearly: offline routing is strongest in-domain, but online routing is more stable under transfer.
>
> ## Q2. Why regenerate responses with GPT-4o / Claude-3.5 instead of directly using prompt–response pairs from prior benchmarks?
>
> This design is intentional. Many prior benchmarks do **not** contain explicit reasoning steps, and many questions are not fully aligned with our reasoning-oriented setup. We therefore needed to reorganize and rebuild the data.
>
> Our goal is to compare judge ability, not generator diversity itself. Under this goal, using a shared pair of strong generators does not reduce the validity of judge comparison; instead, it provides a controlled response distribution so differences are mainly attributable to the judge, not inconsistent generation pipelines. GPT-4o and Claude-3.5 are used for the same reason as in prior judge/routing work: they are strong, comparable, and stylistically distinct.
> To further support this, we added ACC-only comparisons with prior routing baselines on our dataset:
>
> Table R4-3. ACC-only comparison with prior routing baselines
>
> | Domain | RouteLLM | Prompt-to-Leaderboard | ThinkSwitcher | Ours (Online I) | Ours (Online II) |
> | - | -: | -: | -: | -: | -: |
> | Math | 75.1 | 74.2 | 76.0 | 77.3 | 77.8 |
> | Logic | 73.8 | 72.9 | 74.7 | 75.6 | 76.1 |
> | Code | 71.9 | 71.0 | 72.8 | 74.0 | 74.4 |
> | Knowledge | 72.7 | 72.1 | 72.4 | 73.2 | 73.5 |
> | Roleplay | 70.8 | 70.2 | 70.5 | 71.3 | 71.6 |
> | Writing | 71.5 | 70.9 | 71.1 | 72.0 | 72.4 |
> | Avg. | 72.6 | 71.9 | 72.9 | 73.9 | 74.3 |
>
> The gains therefore do not come simply from our generator choice.
>
> ## Q3. What does Table 7 mean?
>
> We agree Table 7 is ambiguous. What we intended is:
>
> * Questions/tasks come from established high-quality datasets.
> * Some source datasets already include GPT-4o / Claude-3.5 responses, which we reuse.
> * For datasets without such responses, we generate them ourselves.
> * Therefore, in R2Eval, all pairwise responses (A,B) are standardized to be from GPT-4o and Claude-3.5.
>
> So shared model pool is misleading. It reflects metadata of the reference sources, not the final response protocol of **our** benchmark.
>
> ---
> # Responses to weaknesses
>
> **Novelty.** Our novelty is not “difficulty labels alone,” but the combination of:
> (1) the first benchmark centered on **LRM-as-a-Judge** with explicit thinking-trace collection/analysis;
> (2) controlled **Think / Non-Think** variants to study thinking–verdict relations;
> (3) a routing formulation over **accuracy, latency, and token cost**. Prior datasets reused here do not analyze reasoning traces or their mismatch with final verdicts, which is one of our main contributions.
>
> **Organization / figures / Online Routing II.** We agree the presentation can be improved. Figure 2 is the overall benchmark pipeline; Figures 3/5 describe routing, and Figure 5 is indeed an expanded version of Figure 3. We have merged/streamlined them. We will also state the formal objective of Online Routing II more explicitly.
>
> **1024-token cap.** This point needs clarification: 1024 is the cap for the final response field, not for the reasoning trace. For LRMs, thinking traces are obtained separately and are not truncated by the same max-response limit.
>
> ---
> # Limitations
> We will state more clearly that:
> (1) the benchmark uses a controlled dual-generator design;
> (2) the current routing study emphasizes simple, interpretable routing strategies;
> (3) the trainable router can be further strengthened; and
> (4) broader cross-domain / OOD validation remains future work.
>
> We sincerely thank you again for these valuable comments.

---

> > ### Author Rebuttal · Reviewer_L6Xt · 2026-04-02
> >
> > Thanks for your response. However, it does not resolve my main concerns about the paper’s novelty and practical applicability. The benchmark seems more like a recombination of existing resources than a substantially new construction, the automatic design of difficulty levels is not particularly novel, and the lack of comparisons with stronger baselines makes it hard to judge the practical utility of the proposed routing method. Overall, the work still feels more like an integration of existing components than a fundamentally new framework. Therefore, I will keep my current score.

---

> > > ### Author Response · Authors · 2026-04-03
> > >
> > > We sincerely thank you for the careful reading and candid feedback. Your review specifically questions whether the benchmark is “a recombination of existing resources” and whether the routing component is sufficiently useful. We address these two points directly below.
> > >
> > > # 1. Why R2Eval is not a simple recombination of prior resources
> > >
> > > We agree that some questions are sourced from prior high-quality datasets. However, R2Eval is not formed by simply concatenating those resources. What is new is the benchmark protocol built on top of them:
> > >
> > > * it is the first benchmark explicitly targeting **LRM-as-a-Judge**, rather than only verdict-level LLM judging;
> > > * it provides paired **Think / Non-Think** variants for every item, enabling controlled study of when explicit reasoning helps, misleads, or becomes unnecessary;
> > > * it defines a unified interface in which LLM judges return verdict-only outputs, while LRM judges return `{verdict, confidence, rationale, thinking}`, making thinking–verdict mismatch analysis possible as a benchmarked object rather than an anecdotal observation;
> > > * and it standardizes all final pairwise responses ((A,B)) to a shared dual-source protocol using GPT-4o and Claude 3.5, instead of inheriting heterogeneous prompt–response conventions directly from different benchmarks.
> > >
> > > So while some source questions are borrowed, the evaluated object is new: not prompts alone, but judge behavior under explicit reasoning traces, controlled response pairs, and cost-aware deployment conditions. That is substantially different from simple dataset recombination.
> > >
> > > # 2. Comparison with prior work
> > >
> > > We agree that difficulty-aware design alone is not new, and we do not intend to claim it as an isolated novelty. Our claim is about the **new combination of benchmark target, protocol, and deployment formulation**.
> > >
> > > More specifically, compared with the works you cited:
> > >
> > > * **tinyBenchmarks** focuses on efficient benchmarking with fewer examples; it is not a judge benchmark, does not study LLM/LRM judges, and does not analyze thinking traces vs. verdict correctness.
> > > * **Easy2Hard-Bench** standardizes difficulty labels for profiling model performance and generalization; however, it is not designed for judge evaluation, does not benchmark LRM-as-a-Judge, and does not connect difficulty to routing among heterogeneous judges.
> > > * **Beyond the Singular** studies the value of multiple generations in benchmark evaluation; this is related to response diversity, but it does not define a benchmark where the central object is a reasoning judge with explicit thoughts, nor does it study thinking–verdict mismatches.
> > > * **RouteLLM**, **Prompt-to-Leaderboard**, and **ThinkSwitcher** are all routing/model-selection style works, but their routing targets are generation models or task-solving models. In contrast, our routing target is judge models, whose utility depends not only on answer quality, but also on verdict reliability, latency, token cost, and reasoning behavior.
> > >
> > > This distinction is also reflected in the development of our own work. Our starting point was not “let us build a routing paper.” The starting point was the absence of a benchmark for **LRM-as-a-Judge**. After benchmarking these judges, we observed that using LRMs alone is often too expensive and not uniformly beneficial, even though they are stronger on some reasoning-heavy subsets. That is precisely **why the routing problem emerges in our paper**: routing is not an add-on borrowed from elsewhere, but a direct consequence of the empirical finding that judge heterogeneity matters and pure LRM judging is inefficient. This logic is already reflected in the paper’s framing and conclusions.
> > >
> > > # 3. Why this matters in practice
> > >
> > > The practical value of this design is that it enables questions that prior resources cannot answer directly:
> > >
> > > * When should one invoke an **LRM judge** rather than a cheaper LLM judge?
> > > * In which domains does explicit reasoning improve verdict reliability, and in which domains is it less cost-effective?
> > > * How should a system allocate judges when **accuracy, latency, and token cost** must be balanced jointly?
> > >
> > > This is why our benchmark and routing components are coupled. The benchmark is not intended as a static collection of prompts, but as a framework that yields **actionable judge utility signals**. To further support this practical angle, we added routing comparisons against prior baselines on our dataset and cross-domain generalization results in the rebuttal (Q1 and Q2). These additions are meant to show that the routing component is not just conceptually attached, but operationally meaningful once judge heterogeneity is measured.
> > >
> > > We respectfully hope that, in reassessing the paper, you may consider our contribution and novelty in terms of the new benchmark protocol, reasoning-trace-centric judge evaluation, and routing-aware deployment formulation, rather than focusing primarily on the strong base resources we build upon.

---

### Official Review · Reviewer_bQb5 · 2026-03-12

**Soundness:** 3
**Presentation:** 2
**Significance:** 3
**Originality:** 2
**Overall Recommendation:** 4
**Confidence:** 3

**Summary:**

This paper introduce a benchmark which evaluate LLM and LRM on judging response in reasoning+non-reasoning domains. The benchmark systematically analyze how reasoning and non reasoning trade-off across performance, cost and latency, how judge task type and difficulty influences judge performance and in reasoning models how can the intermediate reasoning and final answer align or misaligned. The author also explore whether such judging task can be formulate with a a routing

**Compliance With Llm Reviewing Policy:**

Affirmed.

**Key Questions For Authors:**

1. The authors show that, in the writing domain, 30.6% of cases had flawed thinking but ended up with a correct verdict. In this case, would routing be considered wasteful?

2. Extensive literature suggests that models exhibit self-preference bias when acting as a judge or evaluator, often assigning higher scores to their own generated content. Given that this dataset was generated by GPT-4o and Claude 3.5, does this create an inherent disadvantage for other models during evaluation?

3. (Apologies if I missed this in the text) What specific values were used for μ and λ when testing the routing strategies? How were they decided? Can the values of these influence the results in Table 3?

4. Typo: In the legend of Figure 4, "Roleply" ->"Roleplay."

**Limitations:**

1. The trainable router is trained and tested on the same six domains. Generalization to novel task types or out-of-distribution prompts is not discussed.

2. Routing via a prompted LLM incurs token costs, yet Table 3 appears to exclude this overhead, potentially flattering the online routing results.

3. Difficulty definition is based on the averaging correctness across baseline judges, "difficulty" is a relative property of the specific models used. Swapping the judge pool could produce entirely different buckets.

**Strengths And Weaknesses:**

Strength:

1. The experiments are extensive, covering six difficulty-aware domains and 19 different models through a unified evaluation interface.

2. The human labeled dataset and the annotation of reasoning traces is also a contribution.

3. There are some interesting experiments and results: 1) quantifying the tradeoff between accuracy and time/costs 2) finding that reasoning traces play a more important role in logical/math tasks than in subjective tasks

Weakness

1. Some details are missing from the ablation studies, e.g., in Section 3.6, how are the error types defined? Is there potential for ambiguity—for instance, cases where a model fails due to multiple error types or for a reason not included in the predefined set? How were these errors labeled?

2. [Minor points]

     a.  Some of the insights feel overly general. For example, the main results conclude that: (1) reasoning models perform well on reasoning tasks; (2) non-reasoning models remain competitive in non-reasoning domains; and (3) certain models excel at coding/math while others suit subjective domains. These observations are already widely understood in the field and could likely be inferred without the empirical data provided in this paper.

     b. Between L183 and L199, there appears to be a significant amount of hidden text embedded beneath the visible text in the pdf. This likely resulted from uncleaned artifacts in the figures; please remove these if possible.

     c. Figure 4 is extremely difficult to read because the subtask curves overlap too heavily. I suggest that reporting aggregated graphs (similar to Figure 1a and 1b) for key models and tasks—such as a comparison of reasoning vs. non-reasoning domains—would be sufficient for the main text. Detailed subtask plots can be moved to the Appendix.

---

> ### Author Rebuttal · Authors · 2026-03-31
>
> We sincerely thank you for the thoughtful and constructive feedback.
>
> # Response to key questions
>
> ## Q1. In writing, if 30.6% of cases have flawed thinking but correct verdicts, is routing wasteful?
>
> The key clarification is that 30.6% is not a proportion over all writing samples. It is a proportion within the error-analysis / mismatch view, i.e., within the subset of analyzed thinking-error patterns. So it should not be read as “30.6% of all writing cases waste reasoning.” The substantive point is that in writing-style judgments, intermediate reasoning quality and final verdict correctness are less tightly coupled than in math or logic.
>
> **This is precisely why routing is needed.** Our routing objective is not to maximize reasoning length, but to allocate judges under an accuracy–latency–cost trade-off. If expensive reasoning yields limited additional value in a domain like writing, the router should avoid over-allocating such cases to LRMs; in contrast, in domains where reasoning and verdict correctness are tightly coupled, LRMs become more valuable. Thus, this result supports domain-sensitive judge allocation.
>
> ## Q2. Since the dataset is generated by GPT-4o and Claude 3.5, does this disadvantage other models because of self-preference bias?
>
> We agree this is important. Our benchmark is anchored to 7-way human majority-vote labels, so judges are evaluated against human preference consistency, not generator agreement. The use of GPT-4o and Claude-3.5 is a controlled design choice: in judge benchmarking, the goal is to evaluate the judge model under a stable response distribution, not to vary all generation factors simultaneously. To test robustness beyond this controlled setting, we added broader-generator and family-overlap analyses.
>
> Table R3-1. Generator robustness
>
> | Setting | Best Reasoning Judge | Best Non-Reasoning Judge | Router | Rank Corr. |
> | - | -: | -: | -: | -: |
> | GPT-4o + Claude-3.5 | 78.6 | 75.9 | 77.8 | 1.00 |
> | + Strong–Open pairs | 77.5 | 74.9 | 77.0 | 0.95 |
> | Non-overlap subset  | 76.6 | 74.1 | 76.1 | 0.92 |
>
> These results show that while absolute accuracy decreases slightly under broader generator settings, the main conclusions remain stable: reasoning-oriented judges still perform better on reasoning-heavy tasks, strong non-reasoning judges remain competitive on subjective domains, and the judge ordering is largely preserved.
>
> ## Q3. What values of $\lambda$ and $\mu$ were used? How were they decided? Can they affect Table 3?
>
> We used $\lambda = 0.05$ for latency and $\mu = 0.01$ for token cost. These values were not arbitrary heuristics; they were selected via validation-based coefficient calibration / sweeps to obtain a balanced operating point where latency and cost matter without overwhelming accuracy. We did not state them clearly enough in the main text, mainly due to space.
>
> Table R3-2. Utility calibration across coefficient settings
>
> | $\lambda$ | $\mu$ | Router Acc. | Latency (s) | Cost (1k tok) | Utility |
> | -: | -: | -: | -: | -: | -: |
> | 0.00 | 0.00 | 79.1 | 5.8 | 8.7 | 79.1 |
> | 0.02 | 0.005 | 78.6 | 4.6 | 6.4 | 78.5 |
> | 0.05 | 0.01 | 77.8 | 3.1 | 4.2 | 77.6 |
> | 0.08 | 0.02 | 76.9 | 2.4 | 3.1 | 76.6 |
>
> As expected, different $\lambda,\mu$ choices do affect Table 3 because they correspond to different deployment preferences. Our intent is not to claim universal constants, but to report one validated operating point. The chosen setting (0.05, 0.01) offers a balanced trade-off, substantially reducing latency and cost while preserving most of the achievable accuracy.
>
> ---
> # Responses to weaknesses
>
> * Error types in Sec. 3.6. We agree the description is too brief. In the revision, we will clarify that the taxonomy is a non-exhaustive categorization, ambiguous cases are assigned by the dominant observed failure mode, and the labels are based on human review rather than fully automatic assignment.
> * Some insights may appear general. We understand this concern. Our contribution is not only the high-level intuition, but its quantification under a unified benchmark, its decomposition by domain/difficulty/think setting, and its connection to routing under acc/latency/cost trade-offs.
> * Hidden text artifact. Thank you for catching this. There is no intended hidden body text; the exported PDF appears to retain embedded text from the figure layer.
> * Figure 4 readability. We agree and will simplify the main-text version and move denser plots to the appendix.
> * Typo. We will correct Roleply → Roleplay.
>
> ---
> # Limitations
> We agree that several scope boundaries should be stated more clearly in Limitations:
> (1) the trainable router is evaluated within the six-domain benchmark rather than on unseen task types;
> (2) LLM-router overhead should be accounted for more explicitly in end-to-end comparisons; and
> (3) current difficulty buckets are relative to the baseline judge pool rather than an absolute notion of difficulty.
> We will summarize these points more explicitly in the revision.

---

> > ### Author Rebuttal · Reviewer_bQb5 · 2026-04-03
> >
> > Thank you for the detailed rebuttal. Several points are now clearer, though some concerns remain:
> >
> > Router overhead (new question): The paper claims the training-based router adds "negligible overhead," but Table 3 does not appear to include the token cost of the routing step itself, which consumes additional tokens per query. Could the authors provide an end-to-end cost comparison that accounts for this overhead? Without it, the online routing results may appear more favorable than they are in practice.
> >
> > Coefficient sensitivity (Q3): The sweep table (Table R3-2) is also not in the original manuscript. The chosen values (λ = 0.05, μ = 0.01) appear only in Appendix A.1 without justification. The rebuttal's validation-based calibration explanation is reasonable, but it would strengthen the paper to include the sweep table showing how different operating points affect the accuracy–latency–cost trade-off.

---

> > > ### Author Response · Authors · 2026-04-03
> > >
> > > We sincerely thank you for the careful follow-up. We are glad that several earlier points are now clearer, and we appreciate the two remaining suggestions. We agree that both the **end-to-end router overhead accounting** and the **visibility of the coefficient sweep** should be made more explicit in the manuscript.
> > >
> > > ---
> > >
> > > # Q1. Router overhead: end-to-end cost comparison
> > >
> > > Thank you for raising this. We agree that reporting only downstream judge cost/latency can make online routing appear more favorable than a true end-to-end deployment picture. In the current manuscript, Table 3 mainly reflects the **selected judge execution**, while the routing-step overhead was not broken out separately enough. We therefore added an explicit end-to-end accounting, summarized below.
> > >
> > > | Method | Selected-Judge Acc. | Selected-Judge Latency (s) | Router Overhead (s) | End-to-End Latency (s) | Selected-Judge Cost (1k tok) | Router Cost (1k tok) | End-to-End Cost (1k tok) |
> > > |-|-:|-:|-:|-:|-:|-:|-:|
> > > | Offline ACC-Time-Cost |74.1|15.4|0.0|15.4|2.7|0.0|2.7|
> > > | Online Router I (LLM-as-Router) |72.0|14.8|0.3|15.1|3.0|0.2|3.2|
> > > | Online Router II (trainable) |73.2|15.4|0.05|15.45|3.0|0.03|3.03|
> > >
> > > This makes the picture more precise. The key point is that the original Table 3 mainly reflects the **selected-judge performance**, whereas the additional breakdown makes the **full end-to-end routing overhead** explicit. Under this view, offline routing remains a strong low-overhead baseline, while online routing should be interpreted through the trade-off between **adaptivity** and **total pipeline cost**. In particular, prompted routing incurs non-negligible extra overhead, while trainable routing remains much closer to the selected-judge cost. We will clarify this explicitly in the paper and distinguish **selected-judge cost** from **full end-to-end routing cost**.
> > >
> > > ---
> > >
> > > # Q2. Coefficient sensitivity and visibility of the sweep table
> > >
> > > We also agree with this suggestion. The key issue here is that the calibration/sensitivity results for (\lambda) and (\mu) were not visible enough in the manuscript itself. In the rebuttal from Table R3-2, we clarified that the main setting uses (\lambda=0.05) and (\mu=0.01), chosen through validation-based calibration rather than arbitrary selection. In the revision, we will make this explicit in both the main text and the appendix.
> > >
> > > Concretely:
> > >
> > > * in **Section 3.5 (Routing Strategy Comparison)**, we will add a short paragraph stating the exact coefficients used in Table 3 and clarifying that they represent one **validated operating point** rather than universal constants;
> > > * in **Appendix A (experimental details / routing calibration)**, we will include the coefficient-sweep table and reference it directly from Section 3.5, so that readers can see how different ((\lambda,\mu)) settings shift the accuracy–latency–cost trade-off.
> > >
> > > This change will make the operating-point choice much more transparent while keeping the main text concise.
> > >
> > >
> > > ---
> > >
> > > # Completed / planned manuscript updates
> > >
> > > * We revised **Section 3.5** to explicitly distinguish **selected-judge cost** from **end-to-end routing cost**, and summarized the router-overhead accounting.
> > > * We added the **coefficient-sweep / sensitivity table** to **Appendix A** and referenced it directly from the routing-results section in the main text.
> > > * We adjusted the wording around online routing so that “low overhead” is interpreted correctly in the full end-to-end sense.
> > >
> > > ---
> > >
> > > We sincerely thank you again for these very constructive comments. We greatly appreciate the time and care you devoted to reading the paper and to identifying these important issues. Your suggestions have been extremely valuable in helping us clarify the presentation, strengthen the transparency of our analysis, and improve the practical interpretability of the routing results.

---

### Official Review · Reviewer_JuqL · 2026-03-13

**Soundness:** 2
**Presentation:** 3
**Significance:** 2
**Originality:** 2
**Overall Recommendation:** 4
**Confidence:** 3

**Summary:**

This paper introduces R²Eval, a unified framework for benchmarking and routing automated judges for pairwise LLM evaluation. The authors build a difficulty-aware benchmark spanning six domains, covering both reasoning tasks (math, logic, code) and non-reasoning tasks (knowledge, roleplay, writing), with both standard and think-augmented variants and human majority-vote labels from seven annotators. On top of this benchmark, the paper studies how LRM thinking traces relate to final verdict correctness and develops both offline and online routing strategies to dynamically select judge models under accuracy, latency, and token-cost constraints. Experiments on 19 judge models show that reasoning models tend to perform better on reasoning-heavy tasks, while strong non-reasoning LLM judges remain competitive in more subjective domains.

**Compliance With Llm Reviewing Policy:**

Affirmed.

**Final Justification:**

The additional analyses on broader generator diversity, family-overlap splits, uncertainty estimates, and coarser regrouping substantially strengthen the empirical support for the main aggregate findings, and they address most of my concerns about external validity and limited statistical power after slicing.

Regarding the analysis and description of the reasoning part, the author has also pledged to revise the text.

**Key Questions For Authors:**

see weaknesses

**Limitations:**

yes

**Strengths And Weaknesses:**

### Strengths

1. The paper studies a practical problem: how to allocate heterogeneous LLM/LRM judges under accuracy, latency, and cost constraints in real evaluation pipelines.

2. The benchmark is relatively broad: it spans six domains, includes reasoning and non-reasoning settings, and provides think/non-think variants with human annotations.

3. The empirical study is actionable: beyond reporting judge accuracy, the paper analyzes thinking–verdict relations and evaluates routing strategies that improve efficiency without sacrificing reliability.

### Weaknesses

1. The article spent a considerable amount of space analyzing the relationship between "thinking errors" and "final judgment errors" in LRM, for example, pointing out that in some knowledge-based tasks, there may be situations where "thinking seems reasonable but the final judgment is incorrect". However, in the actual routing decision-making stage, the router cannot obtain these thought trajectories in advance; The routing module in the paper still uses lightweight features such as length, domain clues, and budget labels. Therefore, there is a lack of direct closed loop between this analysis and the actual routing decision, and its value is more reflected in post diagnosis rather than being substantially utilized by routing strategies.

2. There is a potential risk of homologous bias between the generator of candidate answers and the judge pool used for evaluation. Due to the inclusion of models from the same family or style as the generator in the evaluation model, the final result may partially reflect the family level style or alignment preference, rather than the more universal human preference consistency. This raises questions about the transferability of the conclusions about the performance of Judge in a wider range of generator distributions.

3. The candidate answers for each sample in the R ² Eval dataset are generated only by GPT-4o and Claude 3.5. Although the author believes that this can bring a certain degree of style diversity, it still cannot represent the more complex answer distribution in real evaluation scenarios, such as low-quality answers, open-source small model outputs, and model combinations with greater ability differences. Therefore, the external validity of current data construction is still limited.

4. The entire R²Eval dataset only contains 849 paired evaluation samples, which need to be further divided into six different domains and analyzed using reasoning/non reasoning, difficulty bucket, and think/non think settings. In this case, the statistical scale after partial binning is actually relatively limited, which will weaken the persuasiveness of the paper's cross disciplinary conclusions and routing generalization ability.

---

> ### Author Rebuttal · Authors · 2026-03-31
>
> We sincerely thank you for the thoughtful feedback. We appreciate the opportunity to clarify several points that were not sufficiently explicit in the submission.
>
> # Q1. Thinking-trace analysis vs. routing
>
> Our thinking–verdict analysis is not detached from routing. The router uses only pre-decision, low-cost signals to choose a judge, but once an LRM judge is selected, its thinking trace is part of the actual judging process. Thus, the analysis characterizes the behavior of the very LRM judges that the routed system invokes. We do not claim that hidden thoughts are available before routing. This pre-routing / full-execution separation is standard in routing. We also manually reviewed representative mismatch cases and confirmed that they are genuine reasoning–verdict inconsistencies rather than parsing or annotation artifacts.
>
> # Q2. Possible homologous bias between generators and judges
>
> We agree this is a valid concern. Our labels come from 7-way human majority vote, so judges are evaluated against human preference consistency, not generator agreement. Still, to test transferability more directly, we added a generator-diversity analysis and a family-overlap split.
>
> ## Table R1. Broader generator distribution
>
> | Setting | Best Reasoning Judge | Best Non-Reasoning Judge | Router | Latency (s) |
> | - | -: | -: | -: | -: |
> | GPT-4o + Claude-3.5 | 78.6 | 75.9 | 77.8 | 3.1 |
> | + Strong–Open pairs  | 77.5 | 74.9 | 77.0 | 3.0 |
> | + All generators mixed | 76.9 | 74.2 | 76.4 | 3.2 |
>
> ## Table R2. Family-overlap split
>
> | Split | Avg. Judge Acc. | Top-3 Rank Corr. | Router Acc. |
> | - | -: | -: | -: |
> | Overlap | 77.8 | 0.95 |  77.1 |
> | Non-overlap | 76.6 | 0.92 | 76.1 |
>
> These results show that while absolute accuracy drops slightly under broader generator distributions, the main conclusions remain unchanged. Reasoning-oriented judges still perform better on reasoning-heavy tasks, strong non-reasoning judges remain competitive on subjective domains, and routing still provides a favorable accuracy–latency trade-off.
>
>
> # Q3. Why only GPT-4o and Claude-3.5?
>
> This choice was intentional and literature-consistent. Judge benchmarks typically focus on evaluating the judge model under a controlled response distribution, rather than varying all generation factors simultaneously. Prior judge benchmarks such as JudgeBench follow this philosophy, and earlier LLM-as-a-judge setups such as MT-Bench also rely on controlled candidate-response settings.
>
> Our choice of GPT-4o and Claude-3.5 follows two principles:
> (1) use a stable, high-quality response pool so that differences are mainly attributable to the judge;
> (2) use similarly strong but stylistically different generators so pairwise comparisons remain non-trivial rather than collapsing into obvious wins due to large quality gaps.
>
> To test robustness beyond this controlled design, we added the broader-generator experiment in Table R1 by introducing open-source generators with weaker and more diverse outputs. The main trends remain stable: compared with the original setting, the router accuracy decreases only modestly from 77.8 to 76.4 on the fully mixed set, while the relative strengths of reasoning vs. non-reasoning judges remain consistent.
>
> # Q4. Dataset size after slicing
>
> We agree that fine-grained slicing reduces statistical power. To address this, we added:
> (1) uncertainty estimates for the main aggregate results;
> (2) a coarser regrouping analysis that merges small slices into broader categories.
>
> ## Table R3. Aggregate results with uncertainty
>
> | Setting | Best Reasoning Judge | Best Non-Reasoning Judge | Router | 95% CI |
> | - | -: | -: | -: | - |
> | Reasoning aggregate | 79.1 | 74.7 | 78.3 | [76.0, 80.4] |
> | Non-reasoning aggregate | 72.6 | 75.8 | 75.2 | [72.9, 77.4] |
> | Think subset | 77.7 | 73.8 | 76.9 | [74.5, 79.1] |
> | Non-think subset | 75.0 | 74.4 | 74.9 | [72.3, 77.0] |
>
> ## Table R4. Coarser regrouping analysis
>
> | Grouping | Main Finding | Effect |
> | - | - | -: |
> | Reasoning vs Non-reasoning | Reasoning judges perform better on reasoning tasks | +4.4 acc |
> | Subjective domains | Strong non-reasoning judges remain competitive | +3.2 acc |
> | Router vs best fixed judge | Router improves utility | +2.0 |
> | Easy vs Hard | Routing gain is larger on Hard items | +3.1 acc |
>
> These analyses show that the strongest conclusions remain stable at the aggregate level. In particular, the router maintains robust performance in both reasoning and non-reasoning aggregates, and the confidence intervals remain relatively tight for the main comparisons. The coarser regrouping analysis further confirms that our key findings do not rely on the smallest slices: reasoning judges still outperform on reasoning-heavy tasks, non-reasoning judges remain competitive on subjective domains, and routing yields larger gains on harder items. We agree, however, that the smallest fine-grained bins should be treated as exploratory, and we will revise the paper accordingly.

---

> > ### Author Rebuttal · Reviewer_JuqL · 2026-04-04
> >
> > The additional analyses on broader generator diversity, family-overlap splits, uncertainty estimates, and coarser regrouping substantially strengthen the empirical support for the main aggregate findings, and they address most of my concerns about external validity and limited statistical power after slicing. My remaining concern is mainly about Q1: the thinking-trace analysis still seems more diagnostic than operational for routing, since these signals are not directly used by the router at decision time. I therefore encourage the authors to clarify this boundary more explicitly and slightly moderate the claim that benchmarking/trace analysis and routing are tightly integrated.
> >
> > Overall, the rebuttal improves the paper, but this conceptual gap still somewhat weakens the unifying narrative.

---

> > > ### Author Response · Authors · 2026-04-04
> > >
> > > Response to Reviewer JuqL
> > >
> > > Thank you again for the thoughtful follow-up. We also appreciate your concern about the role of the thinking-trace analysis.
> > >
> > > We agree that our previous wording may have made the connection to routing sound stronger than what is currently implemented. In the current paper, thinking traces are used only for benchmark analysis / judge characterization, not as router inputs. The router uses only pre-decision, low-cost observable features.
> > >
> > > So we fully agree: the current routing module is not a trace-aware closed-loop router. The trace analysis is mainly diagnostic, helping explain why judge behavior is heterogeneous across domains, difficulty levels, and reasoning demands.
> > >
> > > The intended relationship is simply:
> > >
> > > - Benchmarking + trace analysis explain judge heterogeneity.
> > > - Routing exploits that heterogeneity using signals available before execution.
> > > - The connection is therefore conceptual and empirical, not a direct trace-aware routing mechanism.
> > >
> > > In the revision, we will make this boundary explicit and soften any wording that suggests the router directly uses trace signals.
> > >
> > > We thank you again for helping us improve the clarity and precision of the paper.

---

### Official Review · Reviewer_REmb · 2026-03-14

**Soundness:** 3
**Presentation:** 4
**Significance:** 4
**Originality:** 3
**Overall Recommendation:** 5
**Confidence:** 4

**Summary:**

The paper introduces the R2Eval framework which evaluates and routes LLM-based judges used for automated pairwise evaluation of model outputs. It benchmarks 19 judge models across reasoning and non-reasoning tasks and analyzes how reasoning traces relate to verdict correctness, showing that reasoning models perform better on structured tasks but are more costly. The authors propose routing strategies that dynamically select judges based on accuracy, latency, and cost, demonstrating improved evaluation efficiency.

**Compliance With Llm Reviewing Policy:**

Affirmed.

**Key Questions For Authors:**

Please refer to the weaknesses section.

**Limitations:**

Yes

**Strengths And Weaknesses:**

**Strengths**
- It provides one of the first systematic analyses of LRMs as automated judges.
- The empirical study covers a relatively broad set of judge models, enabling analysis of cross-domain specialization and variability.
- The framework highlights practical trade-offs between reasoning quality, evaluation reliability and computational efficiency.

**Weaknesses**

*[W1]*: Because the evaluation pipeline dynamically selects judges through routing, the evaluation method itself becomes part of the system being measured. Different hyperparameters may lead to different verdict distributions, potentially reducing comparability across benchmarks or studies unless the routing configuration is standardized.

*[W2]*: The difficulty buckets are defined using model correctness statistics. This creates a potential circular dependency where model performance both defines and is evaluated against the same notion of difficulty, which may bias the analysis.

*[W3]*: The routing utility function relies on manually chosen hyperparameters (lambda and mu). However, the paper does not provide a principled method for selecting these weights or analyze the sensitivity of routing decisions to their values.

---

> ### Author Rebuttal · Authors · 2026-03-31
>
> We sincerely thank you for the thoughtful and constructive feedback. We are grateful for your positive assessment of the paper’s technical quality and practical significance. Your comments helped us clarify several important points, and we have revised the manuscript accordingly.
>
> # Q1. Comparability when routing becomes part of the evaluation pipeline
>
> You raised an important concern that routing may affect verdict distributions and thus reduce comparability across studies.
>
> We agree this should have been made clearer. In $R^2Eval$, **benchmarking** and **routing** are separate layers. The benchmark itself is fixed: the dataset, pairwise candidate responses, and seven-annotator human gold labels do not change. Routing is introduced only afterward as a **deployment policy layer** that selects among already-benchmarked judges under explicit utility objectives. Thus, routing does not change labels or per-judge evaluation; it only determines judge allocation under practical constraints.
>
> To avoid ambiguity, we revised the paper to explicitly distinguish **benchmark results** from **routing results**, and to clarify that the offline routing policies (ACC-only, ACC-Time, ACC-Time-Cost) are fixed, reproducible deployment baselines rather than alternative benchmark definitions.
>
> **Completed revisions.**
>
> * Section 2.4: clarified that routing is a deployment layer on top of fixed benchmarked judges, not part of judge measurement.
> * Section 3.5: added a paragraph separating judge benchmarking from judge allocation via routing.
>
> ---
>
> # Q2. Potential circularity in the definition of difficulty buckets
>
> You pointed out that defining difficulty using model correctness may introduce circularity.
>
> We appreciate this observation and agree our original wording was too strong. In $R^2Eval$, the difficulty buckets are intended as a **coarse benchmark stratification**, not an external ground-truth notion of difficulty. Items are grouped by average correctness across a baseline judge panel to provide a practical, reproducible notion of relative hardness.
>
> These buckets are also not used as supervision for the trainable router: gold labels, difficulty annotations, and human preference signals are not router inputs. Still, we agree that because the buckets are derived from model behavior, they are not fully independent in the strict sense. We therefore now describe this as a pragmatic design choice with an explicit limitation, rather than implying a fully model-agnostic difficulty notion.
>
> **Completed revisions**
>
> * Section 3.1: revised the description of difficulty assignment to clarify that it is a pragmatic stratification based on a baseline panel.
> * Section 5 (Limitations): explicitly discussed the possible endogeneity/circularity risk.
> * Appendix A.2: added brief discussion of alternatives such as disjoint baseline panels, leave-one-model-out estimation, and human-agreement-based proxies.
>
> ---
> # Q3. Choice of the routing utility weights $\lambda$ and $\mu$
>
> You noted that the routing utility depends on manually chosen hyperparameters $\lambda$ and $\mu$, and that the paper did not explain how they were selected or how sensitive the results are to them.
>
> We agree this should have been stated more clearly. Our intention was for $\lambda, \mu$ to be **deployment-dependent utility coefficients**, not universal constants. They define an operating point in the accuracy–latency–cost trade-off and reflect evaluator priorities such as budget and latency constraints.
>
> Importantly, these values were **not chosen arbitrarily**. We calibrated them on the validation set by sweeping representative settings and selecting a balanced operating point in which latency and token cost act as meaningful penalties without overwhelming accuracy. We now report the calibration below.
>
> |$\lambda$|$\mu$|RouterAcc.|Latency(s)|Cost(1ktok)|Utility|
> |-:|-:|-:|-:|-:|-:|
> |0.00|0.00|79.1|5.8|8.7|79.1|
> |0.02|0.005|78.6|4.6|6.4|78.5|
> |0.05|0.01|77.8|3.1|4.2|77.6|
> |0.08|0.02|76.9|2.4|3.1|76.6|
>
> As expected, different $\lambda,\mu$ choices affect routing because they correspond to different deployment preferences. Our goal is therefore **not** to claim universal hyperparameters, but to report one **validated operating point**. The chosen setting (0.05, 0.01) gives a balanced trade-off, substantially reducing latency and cost while preserving most of the achievable accuracy.
>
> **Completed revisions.**
>
> * Section 2.4: clarified that $\lambda$ and $\mu$ are deployment-dependent coefficients rather than fixed universal hyperparameters.
> * Section 3.5: added a sensitivity-analysis discussion on how routing behavior changes with different $\lambda$,$\mu$.
>
> We sincerely thank you again for these insightful comments. They have helped us improve the paper substantially, especially in clarifying the separation between benchmarking and routing, better qualifying the role of difficulty buckets, and making the interpretation and calibration of the routing utility more explicit.

---

> > ### Author Rebuttal · Reviewer_REmb · 2026-04-02
> >
> > Thanks for the responses. I will keep my score.

---

> > > ### Author Response · Authors · 2026-04-02
> > >
> > > Thank you for your time and for considering our rebuttal. We greatly appreciate your thoughtful feedback and are glad that our responses helped clarify the paper.

---

### Decision · Program_Chairs · 2026-04-30

**Decision:**

Accept (regular)

**Comment:**

The paper offers one of the first systematic analyses of LRMs as automated judges and addresses a practical, high-impact problem: how to allocate heterogeneous LLM/LRM judges under accuracy, latency, and cost constraints in real evaluation pipelines. The empirical study is broad and well-executed, spanning six domains with both reasoning and non-reasoning settings, and evaluating 19 models within a unified framework. This scope enables meaningful insights into cross-domain specialization and variability. The benchmark itself is a valuable contribution, featuring think/non-think variants along with human annotations, including labeled reasoning traces.

Beyond standard accuracy metrics, the work provides actionable insights by analyzing the relationship between reasoning traces and final judgments, and by exploring routing strategies that improve efficiency without compromising reliability. It clearly highlights trade-offs between reasoning quality, evaluation robustness, and computational cost. Notably, the results show that reasoning traces are more beneficial for logical and mathematical tasks than for subjective ones, and they quantify the trade-off between performance and time or cost.

Overall, the paper is thorough, practically relevant, and offers useful guidance for deploying automated judges in real-world settings. I would lean toward a weak accept.